# The golden mimicry complex uses a wide spectrum of defence to deter a community of predators

**Stano Pekár[1]\*, Lenka Petráková[1], Matthew W Bulbert[2], Martin J Whiting[2], Marie E Herberstein[2]**

[1]Department of Botany and Zoology, Masaryk University, Brno, Czech Republic;
[2]Department of Biological Sciences, Macquarie University, North Ryde, Australia

**Abstract** Mimicry complexes typically consist of multiple species that deter predators using similar anti-predatory signals. Mimics in these complexes are assumed to vary in their level of defence from highly defended through to moderately defended, or not defended at all. Here, we report a new multi-order mimicry complex that includes at least 140 different putative mimics from four arthropod orders including ants, wasps, bugs, tree hoppers and spiders. All members of this mimicry complex are characterised by a conspicuous golden body and an ant *Gestalt*, but vary substantially in their defensive traits. However, they were similarly effective at deterring predators - even mildly defended mimics were rarely eaten by a community of invertebrate and vertebrate predators both in the wild and during staged trials. We propose that despite the predominance of less defended mimics the three predatory guilds avoid the mimics because of the additive influence of the various defensive traits.

\*For correspondence: pekar@sci.
muni.cz

**Competing interests:** The authors declare that no competing interests exist.

## Introduction

Mimicry complexes (i.e. a composite of several Müllerian and Batesian mimics) have long fascinated biologists for the theoretical and empirical challenges they pose. Classic Müllerian mimicry was thought to only include equally defended species (*Müller, 1878*), while Batesian mimicry (*Bates, 1862*) describes undefended species resembling defended models. In reality, this simplistic dichotomy (*sensu Speed, 1999*) represents the end-points of a mimetic spectrum that contains species that vary in their mimetic fidelity and/or their degree of defensiveness. A topic of considerable debate (*Sherratt, 2008*) is the putative influence that moderately defended mimics have on the stability of mimicry rings (i.e. a composite of several similar Müllerian mimics). Discrepancy in protection between two mimetic species may dilute the protection of a more defended species (*Speed, 1999*). Great tits (*Parus major*, Paridae, Passeriformes) for instance, attacked artificial models more often when less defended mimics were frequently encountered and food was restricted (*Rowland et al., 2010*). In direct contrast, associative learning mechanisms predict a mutualistic relationship between moderately and highly defended members of the ring (*Rowland et al., 2007*): wild caught great tits equally avoided four species of true bugs with varying levels of defence but which shared similar warning colours (*Hodová Svádová et al., 2013*).

Müllerian mimicry rings typically consist of a few closely related species (*Symula et al., 2001*; *Williams, 2007*; *Alexandrou et al., 2011*), with the exceptions of the species-rich mutillid wasp (*Wilson et al., 2012*, *2015*), *Heliconius* butterfly (*Mallet and Gilbert, 1995*) and bumblebee (*Williams, 2007*) complexes. These complexes, despite being speciose, are taxonomically limited because their members often belong to only a single family or genus (Lepidoptera or Hymenoptera). The mutillid wasp complex is suspected of including species from other orders but this has not been

**eLife digest** Many animals use bright colours to warn a potential predator that they can defend themselves. Wasps, for instance, are armed with a harmful sting and advertise this fact via their distinctive yellow and black stripes. Predators often learn to heed such warnings and avoid these unpalatable animals in future. As a result, animals that mimic another animal's warning signals can reap the benefit of being left alone by predators even if they are otherwise undefended.

Textbooks on evolution are typically full of different examples of mimicry. However, the specifics of these examples are often poorly understood. Ninety years ago a famous Australian entomologist, Alexander Nicholson, suggested the existence of large groups of mimics in the Australian wildlife. More of these so-called "mimetic complexes" have recently been recognized among several species of insect, but not previously in ants.

Now, Pekár et al. have looked at all known ants and ant-like mimics in Australia and discovered over 140 species that use gold and black colours as a warning signal. Most of the species were ants, but the collection of mimics also includes wasps, spiders, true bugs and insects called treehoppers. Some of the mimics were less palatable than others, and they possessed a range of defences, including spines and foul-tasting chemicals.

Pekár et al. then looked in the guts of 12 species of predators in the wild, and found that very few of them ate the mimics. When mimics were offered to three different predators (specifically a lizard and two species of spider), most avoided the mimics regardless of whether they were palatable or unpalatable. Instead, the predators preferred to eat a spider that was not a member of the group of mimics because it lacked the gold colouration.

Further studies are now needed to continue to document the details of this and other mimetic complexes. For example, this includes revealing how the different defences protect the members of the complex from predators do not use vision to recognize their prey and so cannot see the warning colouration. All this is needed to understand evolutionary processes that have fascinated biologists for decades, and explain how such large mimetic complexes evolved and persisted in spite of the influence of the community of predators.

formally presented (*Wilson et al., 2012*, *2015*). A mathematical model suggests that mimicry complexes should be common in multispecies communities with distantly related taxa (*Beatty et al., 2004*). The general lack of data on multi-order mimicry complexes may represent an observer bias. For example, ants have been largely overlooked as Müllerian mimics, yet they are diverse, typically well defended through a range of defensive measures including a sting, noxious chemicals, spines, a thickened cuticle, mandibles and communal attack (*Hölldobler and Wilson, 1990*). There is also a rich diversity of invertebrate taxa that mimic ants.

Species within Müllerian rings thus far described use the same defensive traits. *Heliconius* butterflies, for instance, rely on their distastefulness while mutillid wasps have a venomous sting. The degree of distastefulness or level of venom toxicity within a mimicry ring is expected to vary among members. Experimental investigations, that started a hundred years ago (e.g. *McAtee, 1912*, *1932*), have tended to focus on a single visually-oriented predatory guild and yet most environments contain a wide range of potential predators of mimics. For example, species that mimic ants may encounter several ant-averse and ant-eating predators (*Pekár et al., 2011*). In some microhabitats, such as leaf litter, the ant-eating predators (which often use chemical cues to recognise prey) and ant-adverse nonvisually-orienting predators are indeed more common than visually-orienting ones, thus exerting strong selection pressure on mimics to signal their unpalatability in sensory modalities other than visual. Thus one of the major outstanding questions regarding the contribution of mimics to the function of mimetic rings is evaluating the success of Müllerian and/or Batesian mimicry by analysing the selection pressure exerted by all predators in a given environment (*Blumstein, 2006*).

Here, we describe a novel diverse and species rich mimicry complex consisting of golden ant, wasp, true bug, treehopper, and spider mimics and we investigate hypotheses concerning their unpalatability and resulting efficacy against predation. Mimetic complexes are often composed of rings – groups of species that draw advantages from similar aposematic signals (*Williams, 2007*;

**Table 1.** List of mimic species belonging to the golden complex arranged according to the order, family (subfamily), and genus (subgenus).

| Order | Family (Subfamily) | Genus (Subgenus) | Species |
|---|---|---|---|
| Hymenoptera | Formicidae (Formicinae) | *Camponotus* | *aeneopilosus* Mayr; *aurocinctus* (Smith); *bigenus* Santschi; *ephippium* (Smith); *fergusoni* McArthur; *nigroaeneus* (Smith); *oxleyi* Forel; *piliventris* (Smith); *setosus* Shattuck and McArthur; *suffusus* (Smith); *tasmani* Forel; *thadeus* Shattuck; *wiederkehri* Forel |
| | | *Polyrhachis (Chariomyrma)* | *appendiculata* Emery; *arcuata* (Le Guillou); *aurea* Mayr; *bedoti* Forel; *constricta* Emery; *cydista* Kohout; *cyrus* Forel; *guerini* Roger; *heinlethii* Forel; *lata* Emery; *obtusa* Emery; *pallescens* Mayr; *schoopae* Forel; *senilis* Forel; *vermiculosa* Mayr |
| | | *Polyrhachis (Hagiomyrma)* | *ammon* (Fabricius); *ammonoeides* Roger; *anderseni* Kohout; *angusta* Forel; *archeri* Kohout; *aurora* Kohout; *brisbanensis* Kohout; *brutella* Kohout; *burwelli* Kohout; *callima* Kohout; *capeyorkensis* Kohout; *conciliata* Kohout; *cracenta* Kohout; *crawleyi* Forel; *darlingtoni* Kohout; *denticulata* Karavaiev; *diversa* Kohout; *dougcooki* Kohout; *electra* Kohout; *elengatula* Kohout; *feehani* Kohout; *hoffmani* Kohout; *melanura* Kohout; *nourlangie* Kohout; *penelope* Forel; *pilbara* Kohout; *placida* Kohout; *seducta* Kohout; *semiaurata* Mayr; *stricta* Kohout; *tanami* Kohout; *tenebra* Kohout; *thusnelda* Forel; *trapezoidea* Mayr; *tubifera* Forel; *unicaria* Kohout; *vernoni* Kohout; *weiri* Kohout |
| | | *Polyrhachis (Hedomyrma)* | *argentosa* Forel; *barretti* Clark; *cleopatra* Forel; *consimilis* Smith; *cupreata* Emery; *daemeli* Mayr; *erato* Forel; *euterpe* Forel; *hermione* Emery; *mjobergi* Forel; *ornata* Mayr; *rufifemur* Forel; *terpsichore* Forel; *thais* Forel |
| | Formicidae (Formicinae) | *Polyrhachis (Myrma)* | *andromache* Roger; *foreli* Kohout; *inusitata* Kohout |
| | | *Polyrhachis (Myrmhopla)* | *dispar* Kohout; *dives* Smith; *reclinata* Emery; *sexspinosa* (Latreille) |
| | | *Polyrhachis (Polyrhachis)* | *bellicosa* Smith |
| | Formicidae (Dolichoderinae) | *Dolichoderus* | *angusticornis* Clark; *clarki* Wheeler; *dentatus* Forel; *doriae* Emery; *extensispinus* Forel; *inferus* Shattuck and Marsden; *niger* Crawley; *rufotibialis* Clark; *scabridus* Roger; *scrobiculatus* (Mayr); *turneri* Forel |
| | | *Iridomyrmex* | *anderseni* Shattuck; *azureus* Viehmeyer; *coeruleus* Heterick and Shattuck; *roseatus* Heterick and Shattuck; *ypsilon* Forel |
| | Formicidae (Myrmeciinae) | *Myrmecia* | *athertonensis* Forel; *auriventris* Mayr; *borealis* Ogata and Taylor; *chrysogaster* (Clark); *cydista* (Clark); *eungellensis* Ogata and Taylor; *fabricii* Ogata and Taylor; *flavicoma* Roger; *fulviculis* Forel; *fulvipes* Roger; *gilberti* Forel; *harderi* Forel; *luteiforceps* Wheeler; *mandibularis* Smith; *michaelseni* Forel; *petiolata* Emery; *piliventris* Smith; *rugosa* Wheeler; *tepperi* Emery; *tridentata* Ogata and Taylor |
| | Formicidae | *Diacamma* | *schoedli* Shattuck and Barnett |
| | (Ponerinae) | *Pachycondyla* | *sublaevis* (Emery) |
| | Mutillidae | *Ephutomorpha* | *aurata* (Fabricius) |
| Hemiptera | Eurymelidae | *Eurymela* | *rubrolimbata* Kirkaldy |
| | Rhyparochromidae | *Daerlac* | *apicalis* (Distant); *cephalotes* (Dallas); *nigricans* Distant |
| Araneae | Salticidae | *Myrmarachne* | *erythrocephala* forma *erato* (L. Koch); *erythrocephala* forma *ornata* (L. Koch); *erythrocephala* forma *daemeli* (L. Koch); *luctuosa* forma *aeneopilosa* (L. Koch); *luctuosa* forma *aurea* (Ceccarelli); *macleayana* forma *foreli* (Bradley) |
| | | *Ligonipes* | *illustris* Karsch |
| | | *Ohilimia* | *scutellata* (Kritscher) |
| | Corinnidae | *Nyssus* | *luteofinis* Raven |
| | Gnaphosidae | *Eilica* | sp. |

*Wilson et al., 2015*). Thus, our first step was the identification of separate rings to understand how the mimicry complex evolved and is reinforced. We therefore tested whether the rings have a sympatric distribution, which would indicate a mutualistic relationship between mimics and whether mimics within a ring share common ancestry.

Our second step was to quantify the relationship between the proposed aposematic signal (golden/black colouration) and the level of noxiousness or distastefulness of the mimics that display

the signal. Aposematic warning signals and distastefulness are expected to evolve together within a species (e.g. *Ruxton et al., 2004*) and thus, the strength of the distasteful stimulus should coincide with the conspicuousness of the warning signal (*Speed and Ruxton, 2007*). In this mimicry complex, we investigated whether there is a positive relationship between signal intensity and unpalatability across species as has recently been found among lady-bird beetles (*María Arenas et al., 2015*). This

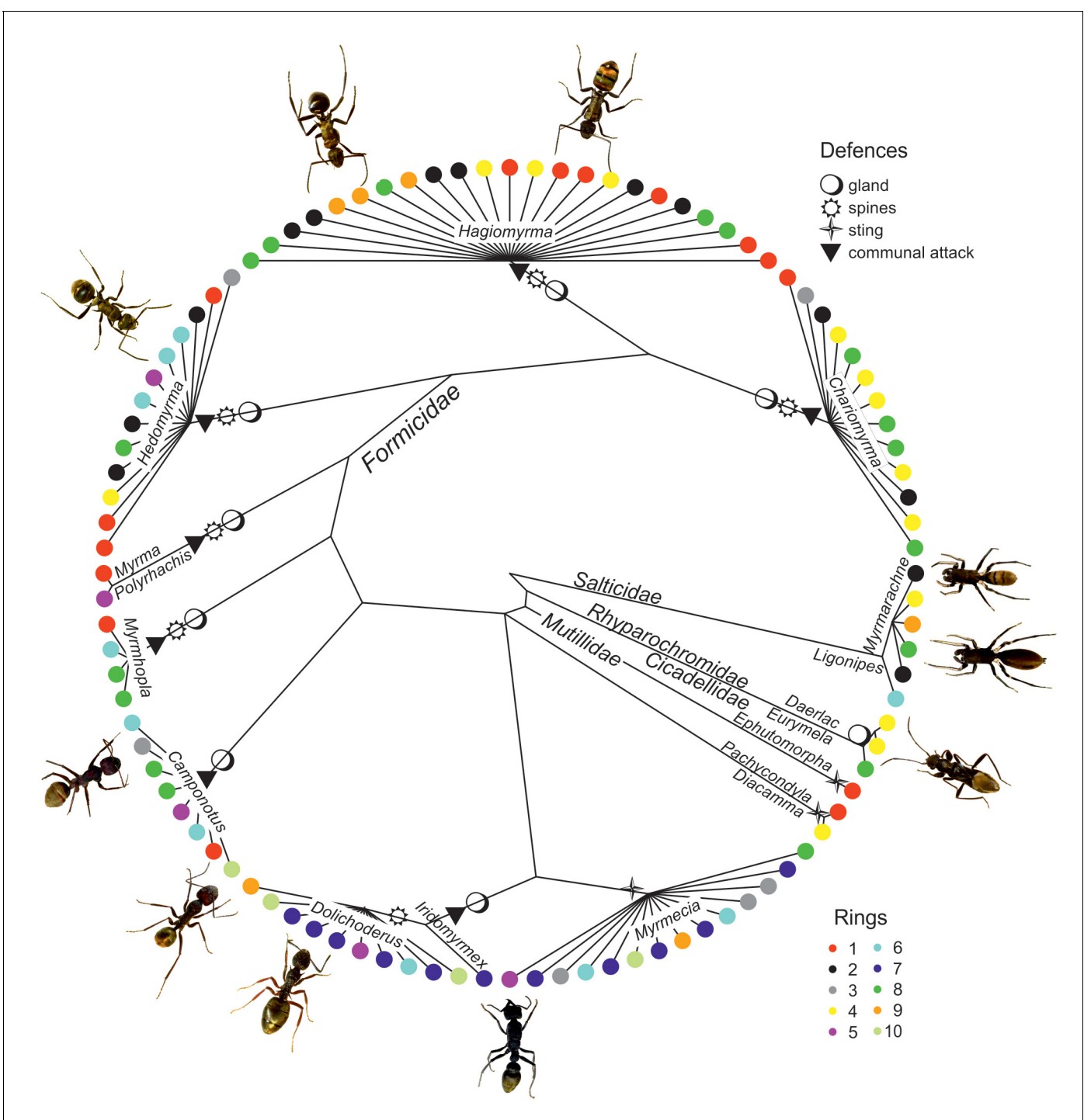

**Figure 1.** Projection of the mimicry complex on a pruned phylogenetic tree. The tips of the tree represent species and full circles are coloured according to putative ring classification. Representatives of each ring are shown. Four types of defence are projected on the branches.

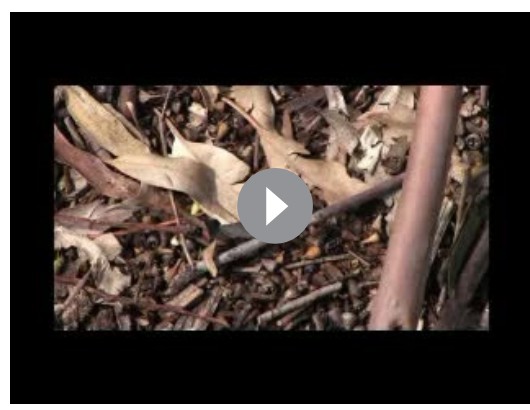

**Video 1.** Overview of selected mimetic species belonging to the golden complex.

would mean that truly Müllerian mimics are expected to honestly signal their distastefulness but truly Batesian mimics display a conspicuous warning signal without the noxious stimulus. We tested these assumptions by correlating the size of the warning signal (golden/back colouration) with a measure of distastefulness derived from the sum of defences found in the mimics.

Our critical third step was to extend the scope of previous studies of mimicry complexes by experimentally testing the efficacy of the aposematic signal, and indeed the validity of the Müllerian mimicry claim, through staged predator-prey trials. Based on the broad diversity of defences used by mimics in this complex, we predict that predators should learn quickly to avoid mimics (*Beatty et al., 2004*), and that mimics should be protected from a range of predators.

Finally, we examined how the aposematic signal and the level of defensiveness interplay with a community of predators that rely on multiple sensory modalities. Mimicry complexes are typically characterised by a distinct colour pattern and hence experiments are often biased towards visual-oriented predators and usually vertebrates. In reality, the members of a complex are exposed to predators that use a variety of sensory modalities and foraging modes to detect and subdue their prey. We combined staged predatory trials with a dietary analysis of wild populations of a community of predators to test the hypothesis that more unpalatable species are attacked less frequently by a community of predators.

## Results

### Description of the mimetic complex

We uncovered a cross-order mimicry complex, consisting of at least 140 arthropod species including 126 ant species, seven spider species (and several forms), three species of true bugs, one mutillid wasp and one treehopper (*Table 1*). Members of the complex are all ant-like in appearance and characterised by a dorsal patch of golden coloration combined with blackish colouration (*Figure 1*, *Video 1*), which we recognise as an aposematic signal (RGB contrast: mean = 241.4, SD = 58.1, N = 55). Related non-mimetic species are typically uniformly black or brown.

The golden sheen is achieved by diverse modes. Among the hymenopteran representatives (ants and wasps), the golden sheen is generated only by pubescence of varying density (*Figure 2*). Consequently, this visual effect entirely disappears when they are submerged in ethanol. In contrast, true bugs and spiders use a combination of hairs and yellow pigmentation (in the exocuticle of the abdomen/gaster), while the treehopper appears to use only pigment as the golden shine is lost after death.

The cluster analysis suggested the existence of ten putative rings within the complex, according to variation in the dorsal colour pattern but the differences were rather small (*Figure 3A*). These rings were distributed across the Australian continent with a considerable overlap. For instance, rings in the east coast of Australia overlapped where species richness was the greatest (*Figure 3B*). However, the degree of sympatry in a ring was not related to colour pattern (estimated from 24 measures of the colour pattern, body size and area of golden patch) similarity (Mantel test, r = −0.03, p=0.79). We did not find a strong effect of phylogeny on the classification into rings (GLS, Pagel's λ = 0.22, *Figure 1*). All this indicates that the splitting into ten putative rings was artificial and there is only one complex.

### Mimicry and degree of unpalatability

We determined the unpalatability of 100 species belonging to the golden mimicry complex. Because several complementary forms of defence are likely, we estimated total unpalatability for each mimic

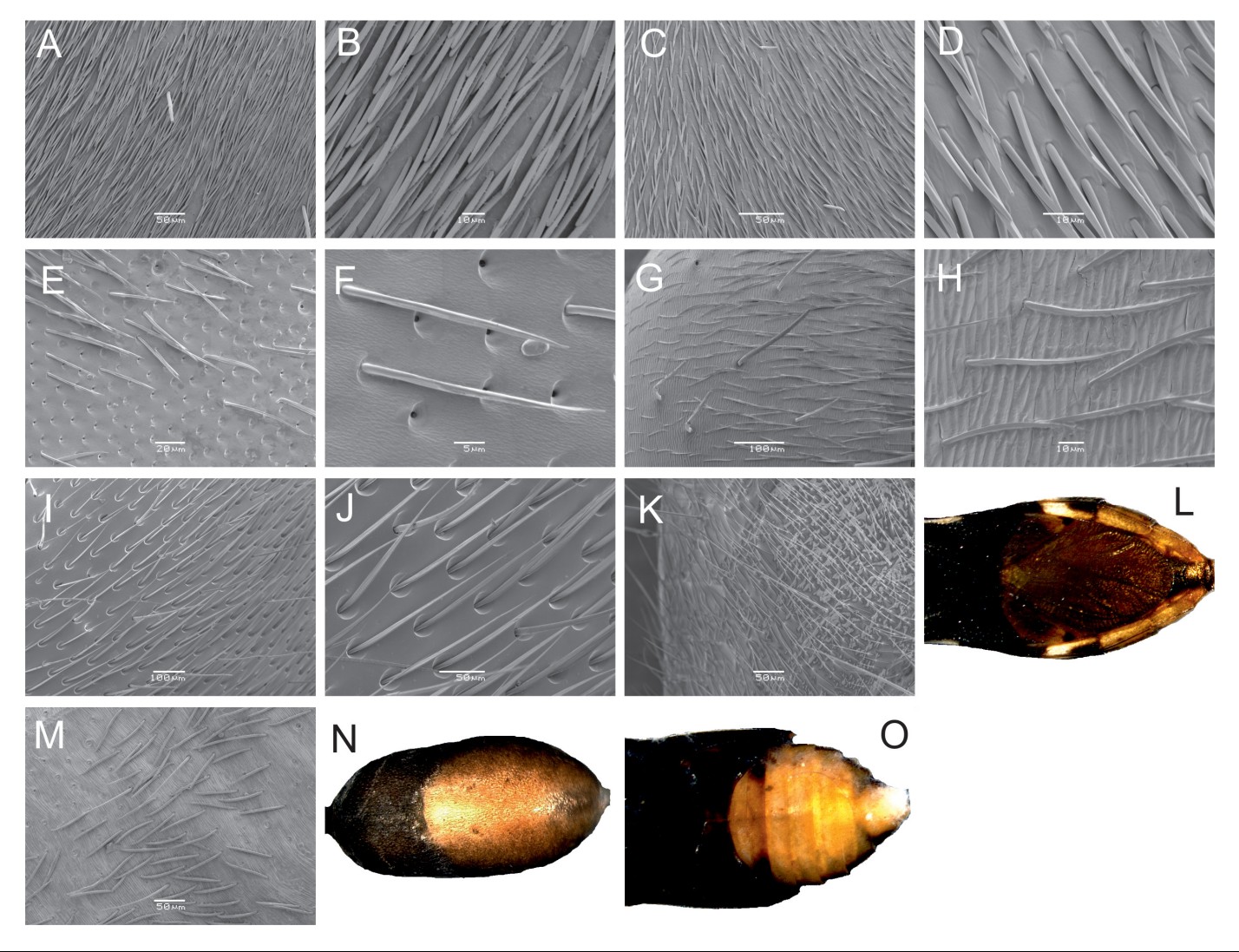

**Figure 2.** Modes of production of the golden shine. (**A**) *Polyrhachis ammon*, gaster, SEM. (**B**) *P. ammon*, detail of hairs. (**C**) *Dolichoderus clarki*, gaster, SEM. (**D**) *D. clarki*, detail of hairs. (**E**) *Myrmecia tepperi*, gaster, SEM. (**F**) *M. tepperi*, detail of hairs. (**G**) *Camponotus aeneopilosus*, gaster, SEM. (**H**) *C. aeneopilosus*, detail of hairs. (**I**) *Ephutomorpha aurata*, gaster, SEM. (**J**) *E. aurata*, detail of hairs. (**K**) *Daerlac nigricans*, gaster, SEM. (**L**) *D. nigricans*, pigment. (**M**) *Myrmarachne luctuosa*, abdomen, SEM. (**N**) *M. luctuosa*, pigment on abdomen. (**O**) *Eurymela rubrolimbata*, pigment on gaster. SEM (**A–K, M**) and light photography (**L, N, O**) of the dorsal side of mimics.

based on the sum of morphological, chemical, and behavioural defensive traits. Our analyses uncovered a spectrum of mimics from non-defended (Batesian) to highly defended (Müllerian) species. For instance, the spider and treehopper species had the lowest unpalatability score (i.e. sum of normalised measurements on defensive traits) and were therefore classified as Batesian mimics. *Daerlac* bugs, mutillid wasps and all ant species were unpalatable, but we found considerable variation in the degree of unpalatability ranging from quasi-Müllerian to unambiguously Müllerian, mimics. The index of unpalatability had an overall normal distribution (Shapiro-Wilk normality test, W = 0.98, p=0.21) with the highest frequency consisting of moderately-palatable species. Unpalatability varied in each ring but the variation was not significantly different among the ten rings suggested previously (Bartlett test, $K^2_6 = 11$, p=0.09, *Figure 4A*) supporting further the existence of only one complex. We discovered that the absolute area of the golden colouration increased with unpalatability (controlling for the effect of phylogeny and body size: GLS, $F_{1,91} = 21.1$, p<0.0001, *Figure 4B*) supporting the hypothesis that the golden colour is an aposematic signal.

## Predation pressure by a community of predators

We quantified predation of mimics by the 12 most abundant predatory species that co-occurred with 13 sympatric mimetic species belonging to the complex (*Table 2*). We analysed gut content or faeces of these predators (553 spiders, 50 skinks, and 48 birds), representing three different guilds, using Next Generation Sequencing, which allows identification of prey to species. We found that individuals of all predatory species were positive for primers targeting the DNA of ants, spiders, and mutillid wasps. The frequency of captured mimics was not significantly different from their availability in the environment ($X^2_{12}$ = 1.5, p=0.99) because mimics were very rare in the field. Only three mimetic species (*Camponotus aeneopilosus*, *Myrmarachne erythrocephala*, *Myrmarachne luctuosa*) were detected in the faeces/gut of predators. Visually-oriented and non-visually oriented eurypha- gous (ant-adverse) predators captured mimics at less than 4% frequency, whereas specialized ant- eating predators captured mostly Batesian mimics (*Table 2*). The overall attack rate estimated for all predators was marginally significantly negatively related to the unpalatability of mimics when it included two Batesian mimics (GAM, $F_{3.8,4.6}$ = 3.8, p=0.045, *Figure 5A*).

## Trials with selected predators

We exposed five species of mimics (ants, spiders and bugs) of varying unpalatability (*Table 3*) and one non-mimic (spider) to one representative of each of three guilds of naïve predator species (eur- yphagous skink, ant-adverse spider, ant specialist spider). Eastern water skinks (N = 26) are visually- oriented generalist predators and captured (i.e. ate) non-mimetic spiders at significantly higher fre- quency than members of the mimetic complex such as ants, spiders and bugs (GEE-b, $X^2_5$ = 125.8, p<0.0001, *Figure 6*). In accordance with our classification of unpalatability, we found a strong posi- tive relationship between prey unpalatability and the post-attack response of skinks (GLM, $F_{1,4}$ = 19.4, p=0.012, *Figure 4C*). Mimic or non-mimic spiders were immediately eaten, but skinks showed an adverse response to mimic ants and bugs. These prey were clearly distasteful because the skinks immediately cleaned their mouths and frequently spat the prey out.

We found similar patterns in the non-visually oriented ant-adverse *Lampona* spiders (N = 25), which captured non-mimic spiders significantly more frequently than all mimics (Cochran test, $X^2_5$ = 104, p<0.0001, *Figure 6*), showing no differentiation between the mimics.

By contrast, the visually oriented ant-eating specialist *Servaea* spiders (N = 27) readily attacked and captured mimics as well as non-mimics. Although these spiders readily attacked the mimics they did appear to differentiate between their levels of defence (attack: GEE-b, $X^2_4$ = 15763, p<0.0001, *Figure 6*). The most unpalatable ant was attacked at a significantly lower frequency than other

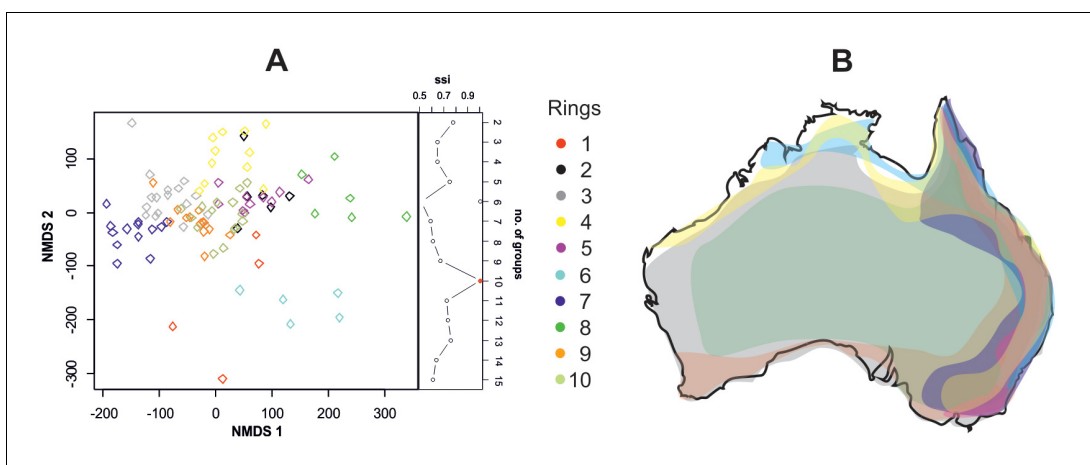

**Figure 3.** Mimetic rings and their distribution. (A) NMDS ordination of species classified into ten putative rings (stress = 0.16). The rings were distinguished by k-means clustering using ssi criterion (in which the maximum value indicates the correct number of clusters), which is shown on the right of the plot. (B) Map of distribution of ten putative rings in Australia.

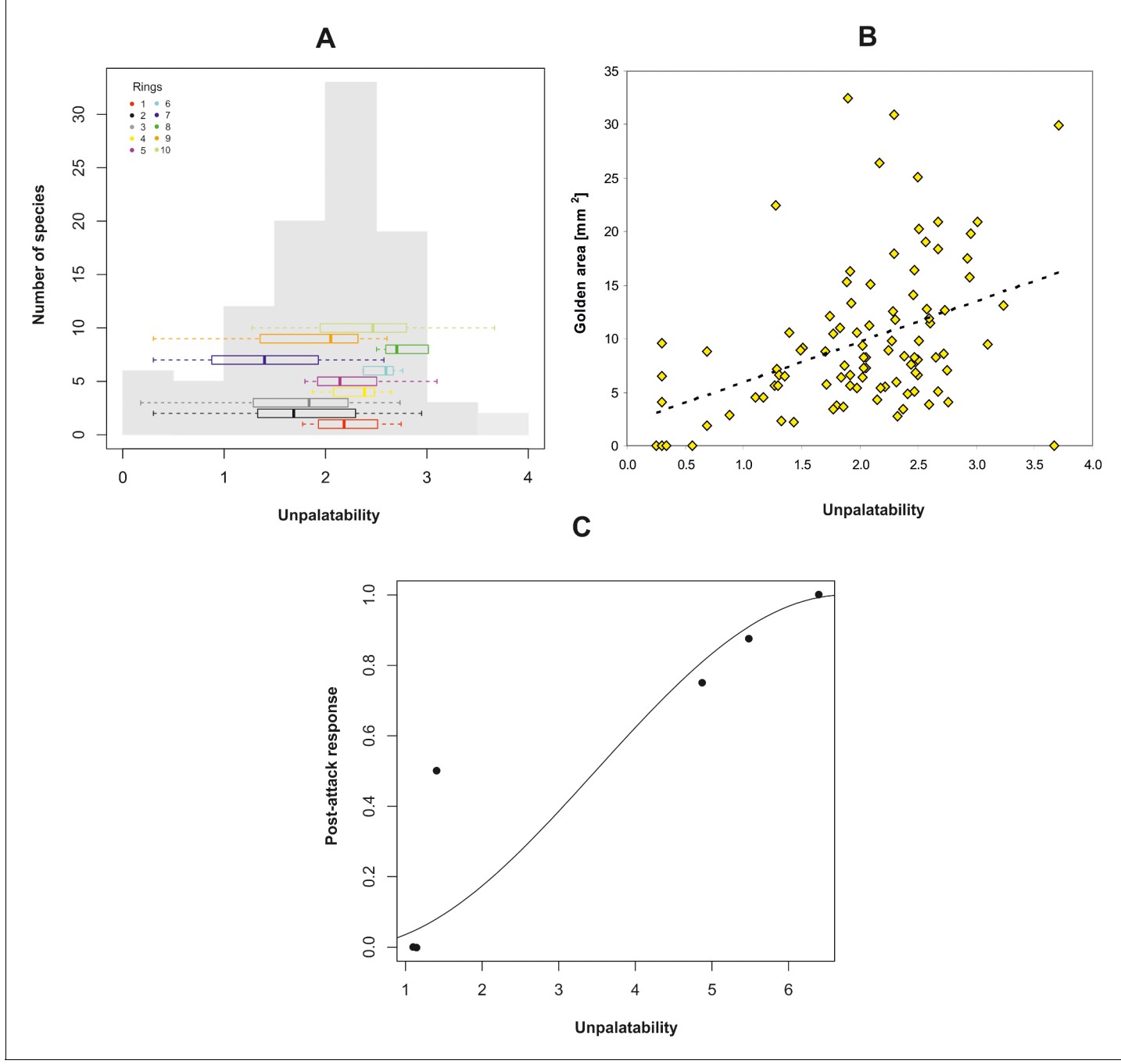

**Figure 4.** Effect of unpalatability. (A) Histogram of unpalatability of the mimetic complex with horizontal boxplots for each ring. (B) Relationship between the absolute area of dorsal golden colouration of mimics and their unpalatability. (C) Relationship between unpalatability and post-attack response of skinks to the six prey species from being eaten (0), to cleaning its mouth (0.5) and spitting out the prey item (1).

mimics (contrasts, p<0.008) and moderately unpalatable ant species were captured at a significantly lower frequency than bug and spider mimics (contrasts, p<0.006).

In summary, euryphagous and ant-adverse predators were less likely to capture a mimic than a non-mimic while the ant-specialist attacked the ants and ant-mimics alike but the likelihood of attack varied according to the mimic's actual unpalatability. There was no relationship between levels of unpalatability and the sum of predation pressure (K-values) exerted by the three predatory guilds (LM, $F_{1,3} = 1.83$, p=0.27, *Figure 5B*).

**Table 2.** The percentage of predator individuals found with DNA (Next Generation Sequencing) of mimics in their gut/faeces. The primer specific for ants amplified on average 9.3% individuals of predators, the primer specific for *Myrmarachne* spiders amplified 3.9% individuals of predators, and the primer specific for mutillids amplified 9.2% individuals of predators. The predators (553 spiders, 50 skinks, 48 birds) were collected on the Macquarie University campus. They are grouped according to their guild. *N* gives the number of individual predators screened. The mimics are arranged from the most to the least palatable (left to right).

| Predator guild Species (Family) | P. ornata | P. vermiculosa | P. aurea | P. ammon | P. erato | M. piliventris | M. tepperi | C. aeneopilosus | E. aurata | D. cephalotes | D. nigricans | M. erythrocephala | M. luctuosa | N |
|---|---|---|---|---|---|---|---|---|---|---|---|---|---|---|
| **Visually-oriented euryphagous** | | | | | | | | | | | | | | |
| Eulamprus quoyii (Scincidae) | 0 | 0 | 0 | 0 | 0 | 0 | 0 | 0 | 0 | 0 | 0 | 0 | 0 | 50 |
| Manorina melanocephala (Meliphagidae) | 0 | 0 | 0 | 0 | 0 | 0 | 0 | 0.02 | 0 | 0 | 0 | 0.04 | 0 | 48 |
| Sandalodes superbus (Salticidae) | 0 | 0 | 0 | 0 | 0 | 0 | 0 | 0 | 0 | 0 | 0 | 0 | 0 | 37 |
| Holoplatys planissima (Salticidae) | 0 | 0 | 0 | 0 | 0 | 0 | 0 | 0 | 0 | 0 | 0 | 0 | 0 | 12 |
| Ocrisiona sp. (Salticidae) | 0 | 0 | 0 | 0 | 0 | 0 | 0 | 0 | 0 | 0 | 0 | 3.8 | 3.8 | 26 |
| **Specialised ant-eating** | | | | | | | | | | | | | | |
| Servaea incana (Salticidae) | 0 | 0 | 0 | 0 | 0 | 0 | 0 | 0 | 0 | 0 | 0 | 0 | 1.8 | 114 |
| Euryopis umbilicata (Theridiidae) | 0 | 0 | 0 | 0 | 0 | 0 | 0 | 0 | 0 | 0 | 0 | 9.9 | 8.6 | 101 |
| Euryopis sp. (Theridiidae) | 0 | 0 | 0 | 0 | 0 | 0 | 0 | 0 | 0 | 0 | 0 | 0 | 0 | 12 |
| Hemicloea sp. 1 (Gnaphosidae) | 0 | 0 | 0 | 0 | 0 | 0 | 0 | 5.0 | 0 | 0 | 0 | 0 | 0 | 20 |
| Hemicloea sp. 2 (Gnaphosidae) | 0 | 0 | 0 | 0 | 0 | 0 | 0 | 2.4 | 0 | 0 | 0 | 2.4 | 2.4 | 42 |
| **Non-visually oriented euryphagous** | | | | | | | | | | | | | | |
| Lampona murina (Lamponidae) | 0 | 0 | 0 | 0 | 0 | 0 | 0 | 0 | 0 | 0 | 0 | 2.2 | 4.3 | 46 |
| Clubionia robusta (Clubionidae) | 0 | 0 | 0 | 0 | 0 | 0 | 0 | 0 | 0 | 0 | 0 | 0 | 1.7 | 58 |
| Clubiona sp. (Clubionidae) | 0 | 0 | 0 | 0 | 0 | 0 | 0 | 0 | 0 | 0 | 0 | 2.4 | 2.4 | 85 |

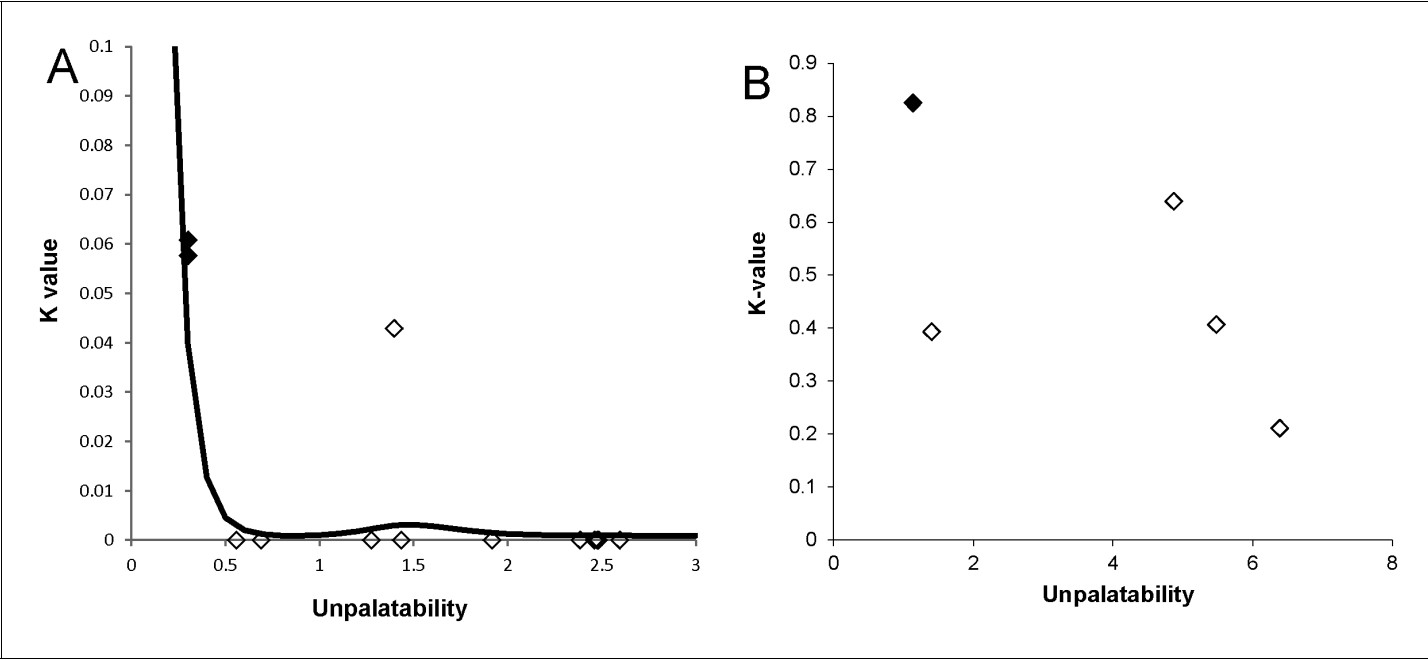

**Figure 5.** Relationship between unpalatability and predation pressure. (**A**) Relationship between unpalatability and the total predation pressure (K-value) on 13 mimics by three guilds of predators (visually-oriented euryphagous, specialized ant-eating, non-visually-oriented ant-adverse). Estimated non-parametric regression model (GAM) is displayed. See *Table 2* for unpalatability of mimics. (**B**) Relationship between unpalatability of five mimics and total predation pressure (K-value) from three representatives of predators (see *Table 3* for unpalatability of mimics). A high K-value indicates high predation while a low K-value indicates low predation. Solid symbols = Batesian mimics, hollow symbols = Müllerian mimics.

## Discussion

Ants have not been considered in Müllerian mimetic rings, which is perhaps surprising given their diversity and the number of known ant-mimics. The lack of ant fauna in mimetic rings possibly reflects an overall lack of conspicuous defensive coloration in ants. However, our study supports a species rich mimicry complex of golden coloured ant models and mimics, adding to the small number of documented large-scale complexes. Mimetic complexes (among several families of beetles) were first recognised by *Nicholson (1927)* for Australian insect fauna. Such complexes are presumably far more widespread than appreciated, but have only been subject to more intensive study relatively recently. Many more large-scale mimetic complexes among ants and other animals (particularly arthropods) are likely to be described in future.

The species richness of this golden complex exceeds those of fish or millipedes (*Marek and Bond, 2009*; *Alexandrou et al., 2011*) but is currently smaller than in mutillid wasp (*Wilson et al.,*

**Table 3.** A list of traits used to assess the unpalatability of five mimics. Values are means (±SE) estimated from 10 measurements. The species are arranged from the most to the least palatable.

| Species | Frequency of biting | Spray chemicals | Number of spines | Total spines length [mm] | Cuticle thickness [mm] | Total body size [mm] | Mandible size [mm] | Gland size [mm²] |
|---|---|---|---|---|---|---|---|---|
| *P. ammon* | 0.1 | 1 | 4 | 3.54 (0.08) | 0.04 (0.002) | 9.04 (0.08) | 1.00 (0.02) | 2.84 (0.16) |
| *P. vermiculosa* | 0 | 1 | 6 | 2.96 (0.08) | 0.03 (0.002) | 5.98 (0.05) | 0.71 (0.03) | 1.62 (0.06) |
| *C. aeneopilosus* | 0.3 | 1 | 0 | 0 | 0.02 (0.001) | 8.04 (0.17) | 0.84 (0.02) | 2.43 (0.09) |
| *D. nigricans* | 0 | 0 | 0 | 0 | 0.02 (0.0002) | 7.40 (0.14) | 0 | 0.57 (0.03) |
| *M. luctuosa* | 0 | 0 | 0 | 0 | 0.02 (0.001) | 7.04 (0.28) | 0 | 0 |

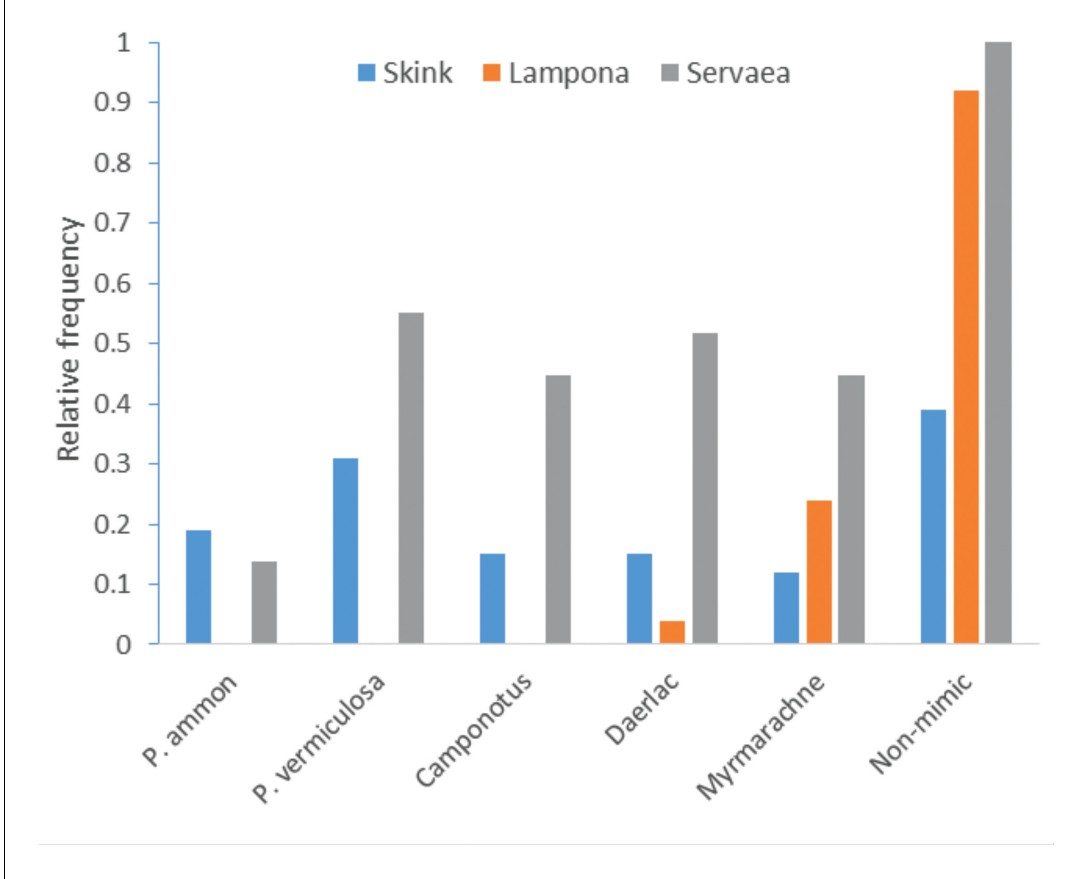

**Figure 6.** Capture of mimics by three predators. Comparison of frequency of attacks on five mimics and one non-mimic by skinks (euryphagous visually-oriented predator); *Lampona* spiders (ant-adverse non-visually oriented predators), and *Servaea* spiders (specialised ant-eating predators).

*2012*, *2015*), heliconid butterfly (*Mallet and Gilbert, 1995*) and bumblebee (*Williams, 2007*) complexes. The key members in the golden complex are ants but the complex is taxonomically broad and includes species with distant phylogenetic origins. Beside the species mentioned, it may include a few other groups, such as bees, flies, and cerambycid beetles, in which some species also have a golden colouration. This is in contrast to complexes of similar scale that only include members from a single order.

The distribution of the majority of the ten putative mimetic rings in the complex overlapped with at least one other ring. Although the cluster analysis distinguished rings, the distances between the rings were not large suggesting that the entire complex is just a single ring. Our findings are not unlike those from other complexes that include rings of different colour pattern, which overlap in geographic distribution (*Williams, 2007*; *Wilson et al., 2012*, *2015*). An overlap of golden rings or actually their non-existence potentially suggests that the variable golden colour patterns produce similar aposematic signals.

Large mimetic complexes include taxa of close and distant phylogenetic origins (e.g., *Linsley et al., 1961*; *Rodriguez et al., 2014*) in which the colour pattern is not only a product of common ancestry but also of convergence. A shared ancestry is apparent for two species of *Myrmarachne* spiders (*Pekár et al., 2017*), three species of *Daerlac* bugs (*Casis and Symonds, 2012*), and several species of *Polyrhachis* ants (*Robson et al., 2015*). The majority though, including *Camponotus*, *Diacamma*, *Pachycondyla*, and *Iridomyrmex* ants, other spider species, *Eurymela* treehopers, and *Ephutomorpha* wasps, have converged on the same phenotype as evidenced by their isolation within lineages of non-mimetic colour patterns. The diversity and geographic pattern of the golden

phenotype is likely a result of co-divergence as found in other complexes (e.g., *Rodriguez et al., 2014*).

Species in Müllerian rings described to date, e.g. heliconid butterflies, use a single defensive trait to teach predators that they are distasteful (e.g., *Van Zandt Brower, 1958*). The lesser-defended individuals in this context are less distasteful. Species in our golden complex use multiple defensive traits including a variety of structural and chemical defences. To understand how these traits interplay with the aposematic signal we created an index that summed all traits known to serve a defensive function. We acknowledge this approach is somewhat simplistic as various traits are differentially effective against various predators. For example, a thick cuticle is effective against arthropod predators but ineffective against vertebrates (*Evans and Sanson, 2005*). On the other hand, spines are effective against vertebrates but ineffective in the defence against arthropod predators (*Mikolajewski et al., 2006*). Different species of closely related predators may also use a variety of adaptations to deal with the defences of prey. For example, some spiders are able to catch ants even with a thick cuticle while others are not (*Pekár and Toft, 2015*). A true index of unpalatability requires an estimation of the total effectiveness of a combination of defensive structures, such as spines and thick cuticle and the toxicity of chemical compounds for each predator-prey combination. Estimates including spines and cuticle thickness may be relatively straight forward but information on the toxicity of chemical compounds is largely not available (see http://toxnet.nlm.nih.gov). Although our approach is simplistic, the positive relationship between the index of unpalatability and the response of skinks strongly suggests that the measure is a good proxy for unpalatability.

Some of the defensive traits were strongly correlated with body size. Body size alone is an anatomical constraint that may prevent predation, thus it is difficult to separate the effect of pure body size from the relative size of some defensive traits (e.g., spines). A strong correlation between body size and the size of defensive traits is natural, because larger spines, for example, require a larger body to support them. It is, however, the absolute size of defences not their relative size that is ecologically important (e.g., a skink will swallow larger spines with greater difficulty than smaller ones). Whatever their single effect is, all these traits together, add to an overall unpalatability of mimics.

Our study describes a new complex with a golden/black colour combination. This combination is somewhat similar to the classic aposematic yellow/black phenotype (*Ruxton et al., 2004*; *Williams, 2007*). Accordingly, our findings support the idea the coverage of golden colour acts as an honest signal of aposematism. All the predators in staged trials exhibited an aversion to the prey with the greatest coverage of golden colour. Furthermore, golden individuals were rarely found in the diet of natural populations of predators.

Honest signalling is further supported by the positive correlation between the extent of the golden colour and unpalatability, controlling for body size in the analyses via phylogenetic correlation or as a covariate. The theory of honest signalling, however, predicts such a relationship within rather than across species. However, recent evidence shows that this relationship can also exist across closely related aposematic species (*Cortesi and Cheney, 2010*; *María Arenas et al., 2015*). It remains to be tested what speciation process and selection forces have driven this pattern. Given the fact, that the complex is composed of many unrelated taxonomic groups that occur in different micro-habitats, the processes are likely to vary. It is necessary to investigate whether such a relationship between species holds for other known mimetic complexes.

Unlike the classic yellow/black combination, members of the golden complex are defined by a metallic golden sheen. This sheen reflects brightly under direct sunlight making the animals very conspicuous, at least to a human observer. Just how important this sheen is in deterring a predator over and above just a matt colour has yet to be determined for this system. In other systems though, reflective iridescence has a significant additive effect (e.g. *Fabricant et al., 2014*) so it is likely that the sheen enhances the detectability of the signal. A possible additional benefit, or even an alternative explanation, is that the sheen is important for reflecting harmful radiation and dissipating heat (*Shi et al., 2015*). Irrespective of the explanation the importance of this colour variant, further experimental investigation is warranted.

The aversion to prey with golden colour was apparent irrespective of the predator's visual sensitivity. Visually- and non-visually orienting predators equally avoided the golden prey suggesting that golden mimics advertise their unpalatability via multiple sensory modalities. The non-visual predator still proportionally avoided the mimics despite not being able to detect the visual signal. This is an area of promising future investigation. Some non-visual predators potentially recognize mimics via

the patterns of vibrations generated by their movements (*Pekár et al., 2011*). Predators typically use multisensory modalities to identify prey (*Bro-Jørgensen, 2010*) such as movement, shape and defensive chemicals they emit.

Our measure of unpalatability suggested that the highest proportion of individuals was moderately defended and defensiveness is distributed from highly defended to weakly defended. This is in agreement with previous claims that mimicry rings are dominated by moderately defended species (*Sherratt, 2008*). With a predominance of moderately defended prey, there is an expectation that their presence will erode the effectiveness of the model. Our findings, based on a limited number of mimics, multiple predatory guilds when taking into account, did not support this notion. Instead, all mimics investigated in our predator-prey trials experienced significantly lower mortality than the non-mimetic species, and this was irrespective of the coverage of gold. We found the skinks were more likely to attack the lesser-defended individuals but when the suite of predators was taken into account there was little variation in the overall predation pressure. This is potentially because some predators may not use the visual signal as a reference and are cuing in on other signals.

Diet analyses of several potential predators also showed predators of all guilds, including non-visually oriented predators, captured mimics with very low frequency. The predators that we used are known to feed on non-mimetic prey too (*Jackson and Harding, 1982*; *Daniels, 1987*; *Platnick, 2000*; *Higgins et al., 2001*; *McGinley et al., 2015*). Indeed, they fed on prey related to the mimics as evidenced by the positive results of designed primers targeting the mimics. We did not measure the availability of alternative prey so we do not have an absolute indication of relative encounter rates. However, birds learn to avoid both Müllerian models and mimics irrespective of the availability of alternative prey (*Lindström et al., 2004*). In addition, there are multiple species of ant and ant models distributed across the sampling habitat occupying both terrestrial and arboreal habitats. The majority of the species in the complex are also ants and ants are typically super abundant due to their sociality. So it seems likely the predators would see these animals on a regular basis.

Using a multi-disciplinary approach here, we faced several constraints. For example, measurements of mimic colour were based on photographs as we were unable to collect all mimics. To correct for inherent differences in illumination used during photographing mimics, the obtained RGB values were standardized but not linearised, and thus should be interpreted with caution. Similarly, the results of experiments with skinks must be interpreted with caution due to constrained randomisation used (Müllerian mimics first). The response of skinks could be affected by the constrained prey order (due to some kind of habituation). Yet, other predators used showed similar response to mimics as skinks, though the order of offered prey was fully randomised.

Here we present the first account of the golden mimetic complex, which we believe is an exciting model for future research. Our work to date is mainly descriptive, and it requires more explicit experimental approaches to generate robust conclusions about the evolution of this golden aposematic signal. For example, the relationship between the level of unpalatability and the quality of the golden signals needs rigorous testing with a greater number of mimics. This would also help in deciphering the role of less-defended mimics in the protection of the entire complex. Similarly, exploring the function of non-visual signals in deterring predators will elucidate how this mimicry ring works.

In conclusion, we present the first account of the golden mimicry complex. It consists of species that show considerable variation in their level and mode of defence, ranging from undefended Batesian mimics to highly defended Müllerian mimics, with most members of the complex somewhere in between. Our data suggest that all members of the golden mimicry complex receive quite similar net protection from a variety of predators presumably because they have a variety of defensive traits. We predict that mimetic complexes that incorporate a range of different forms of defence are not only more common than previously believed, but also more likely result in mutualism among the unequally defended members of the complex.

## Materials and methods

### Description of the mimetic complex

Arthropod species forming the complex were selected based on the presence of a golden colour pattern on the dorsal side of their body and body size falling within the range 3–15 mm. All Australian ants, and known ant-like mimics, were examined for a dorsal golden appearance. The specimens

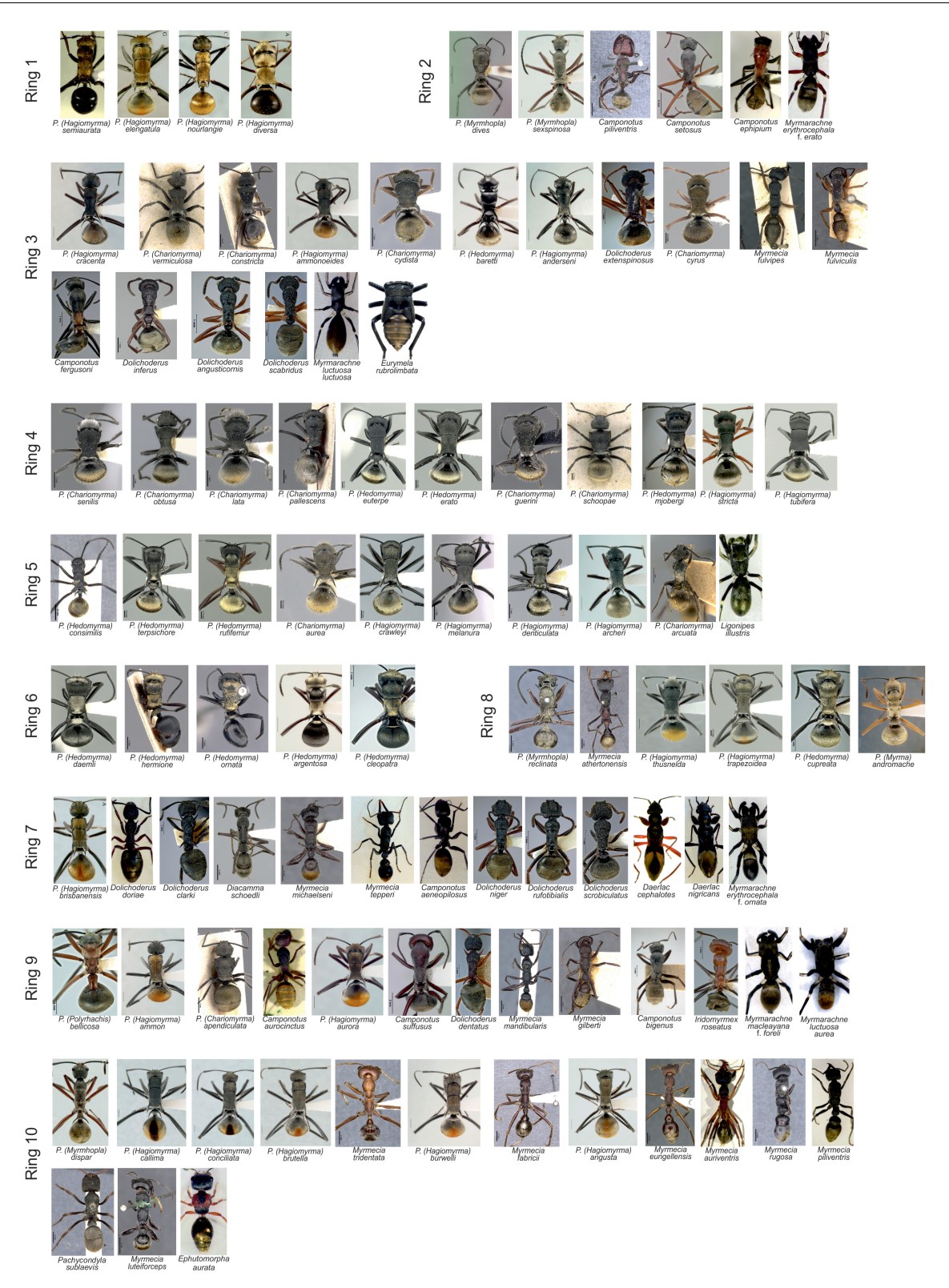

**Figure 7.** Selected list of 100 species representing the ten putative mimetic rings. Species are arranged within each ring from the most to the least unpalatable (left to right). Some pictures of ants are displayed with the permission of AntWiki and R. Kohout.

we examined were either field collected by the authors (Sydney, Blue Mountains, NSW outback, NSW eastern coast, Queensland eastern coast) or from museum collections (Australian Museum, Sydney). All other specimens we viewed were images from AntWiki (www.antwiki.org), Atlas of Living Australia (www.ala.org.au), or taxonomic papers (*Ogata and Taylor, 1991*; *Shattuck and Barnett, 2006*; *McArthur, 2007*; *Kohout, 2010a, 2010b, 2012, 2013a, 2013b*; *Heterick and Shattuck, 2011*; *Casis and Symonds, 2012*; *Shattuck and Marsden, 2013*).

The dorsal colour pattern was estimated from digital images of specimens (*Figure 7*) as performed in other studies on mimetic complexes (*Wilson et al., 2012, 2015*; *Rodriguez et al., 2014*). Field collected specimens (of 41 species) were killed by ethylacetate and arranged in their natural body position. Photographs were taken with a Canon Legria HF G10 against a white background and lit from above by two full spectrum light bulbs (Repti Glo 2.0 UVB, Eko Terra). Photographs of the remaining species (59 species) were obtained mostly from the personal collection of R. Kohout, followed by AntWiki, and Atlas of Living Australia. Prior to taking measurements all photographs were calibrated for the white and/or black colour within ImageJ to correct for different lighting conditions. The average value of each RGB colour component was estimated on the following body parts: dorsal side of head, appendages (femur II), pronotum, mesonotum, metanotum, petiole, first segment of the gaster, and the remaining segments of the gaster/abdomen in ants and wasps using ImageJ. Likewise, for hemipterans and spiders the average colour value of each RGB colour component of corresponding body parts (to that of ants) were examined including the pedipalps/chelicera, frontal third of prosoma, second third and last third of the prosoma, legs (femur III), the first half of the abdomen and the second half of the abdomen. In each species, we also measured the length of each body part (listed above) and an absolute area of the golden colouration on the dorsal side using ImageJ.

To assess colour pattern similarity among mimics and to find how many rings there are, we subjected 24 measures of the colour pattern, body size and area of golden patch of 100 mimics (*Pekár et al., 2016*) to the Non-Metric Multidimensional Scaling based on Bray-Curtis distance matrix. Because various body parts have different area and are thus differently conspicuous to predators, we used weighting of the measures by their relative size. Then we estimated the number of putative rings via k-means clustering using the vegan R package (*Oksanen et al., 2013*). This method used the simple structure index (ssi) as a partition criterion (*Legendre and Legendre, 1998*) as AIC is not available. Red Green Blue (RBG) contrast was estimated using 55 species of mimics. It was computed as the sum of differences in red, green, and blue values representing the golden and black colouration found on the body of mimics.

The phylogenetic relationship at the genus level among the golden mimics was reconstructed from a combination of published phylogenies (*Moreau et al., 2006*; *Regier et al., 2010*; *Robson et al., 2015*; *Pekár et al., 2017*). If species level phylogeny was not available, we used polytomic splitting at the species level. To test for phylogenetic inertia of classification to putative rings we used generalised least squares (GLS) with Pagel's phylogenetic correlation structure (*Paradis, 2006*), in which low value of λ indicates low phylogenetic signal.

The mode by which the animals generated the golden colour (pigments or structural colours) was determined via microscopy. One specimen of each species from each genus (*Ephutomorpha aurata*, *Camponotus aeneopilosus*, *Daerlac nigricans*, *Dolichoderus doriae*, *Polyrhachis ammon*, *P. vermiculosa*, *Ligonipes illustris*, *Myrmarachne luctuosa*, *Myrmecia tepperi*, *Eurymela rubrolimbata*) was further examined via SEM. The golden gaster/abdomen was dried at a room temperature for 10 days, then mounted on a stub and coated with gold prior to viewing under a SEM Jeol JSM 648OLA. The presence of pigments on the gaster/abdomen was examined in ethanol-preserved specimens under Olympus stereomicroscope SZX9 and photographed with a digital camera mounted on the stereomicroscope.

The distribution of mimics across Australia was estimated using occurrence records from collections made by the authors, taxonomic papers (*Ogata and Taylor, 1991*; *Shattuck and Barnett, 2006*; *McArthur, 2007*; *Kohout, 2010a, 2010b, 2012, 2013a, 2013b*; *Heterick and Shattuck, 2011*; *Casis and Symonds, 2012*; *Shattuck and Marsden, 2013*), museum material deposited in the Australian Museum and Queensland Museum and records stored in the database of the Atlas of Living Australia. To test for an association between colour similarity and distribution overlap of species in each mimicry-ring in Australia we used a grid cell of 400 × 400 km. Within each grid occurrence of a species was classified as 1 (if present) or 0 (if not recorded) (*Pekár et al., 2016*). Then a distance

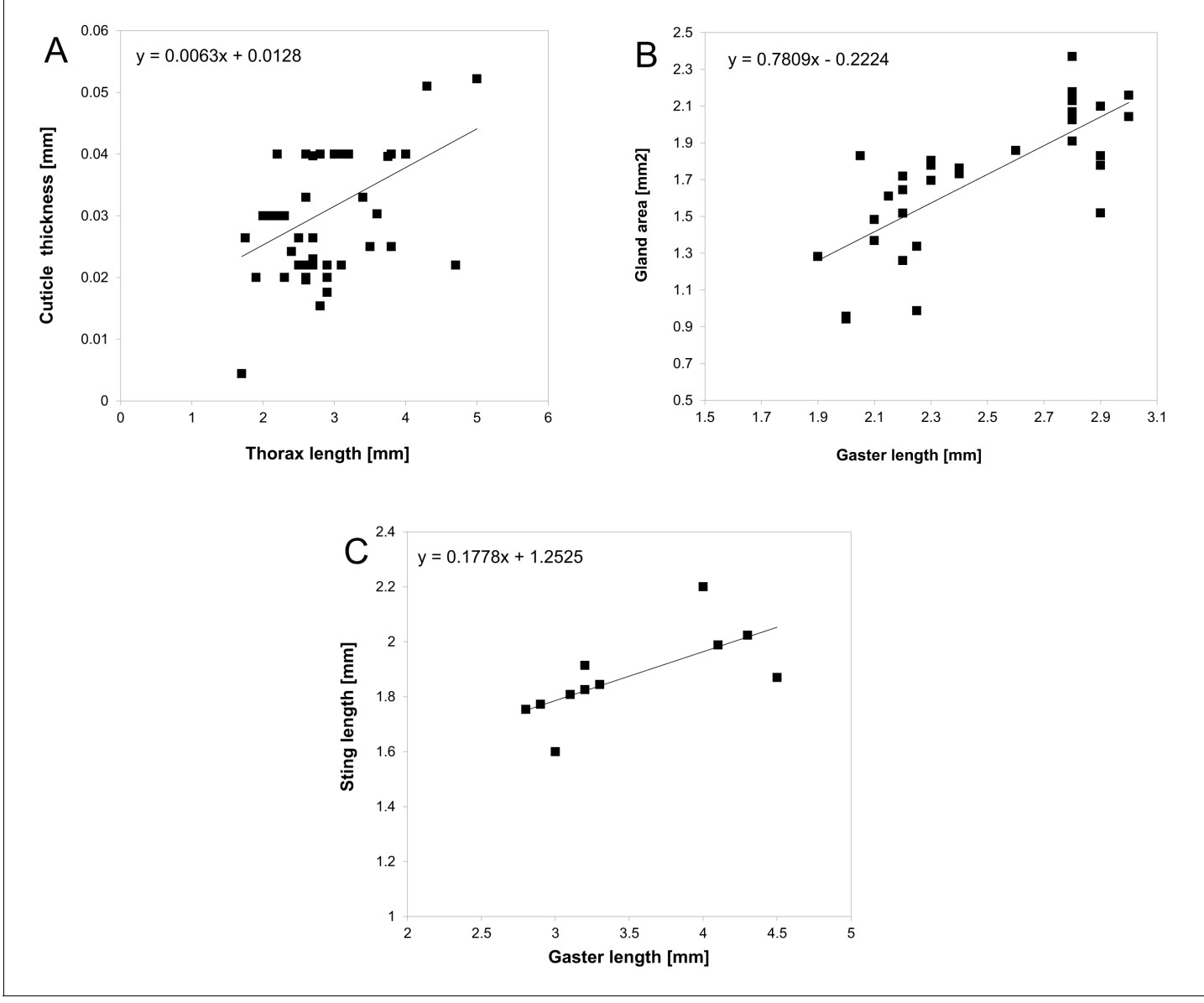

**Figure 8.** Relationship between cuticle thickness and thorax length. (**A**) poison/pygidial gland area and gaster length (**B**) sting length and gaster length (**C**) with estimated linear regression models.

matrix was created from the data using the Bray-Curtis method (*Legendre and Legendre, 1998*). Using the colour pattern data, we created a matrix and generated a distance matrix between species using Euclidean distance. To test a hypothesis that putative rings overlap in their geographic distribution, we studied the association between the geographical occurrence of mimics and colour similarity matrices by means of a Mantel test based on the Pearson correlation coefficient.

## Mimetic unpalatability

We estimated unpalatability in 100 mimics (workers and adults) from the complex using the following seven traits ($x_i$): length of the sting, number of spines, total length of spines, size of mandibles, cuticle thickness, size of poison/pygidial gland, and performance of a communal attack (*Pekár et al., 2016*). Size, number of spines and size of mandibles were measured in one individual from each species. Cuticle thickness and size of the poison/pygidial gland could only be measured from freshly collected specimens. So we collected the mimics (workers in the case of ants) from a variety of places

across Australia (Cairns, Townsville, Brisbane, Sydney, Blue Mountains, Broken Hill, Hay, Macquarie Port, Tweed Heads, Sunshine Coast, Hervey Bay, Agnes Water). In total, 75 specimens belonging to 41 species, eight genera, and four subfamilies were collected alive. Individuals were killed by freezing at −20°C. Cuticle thickness was measured by slicing the thorax (mesosoma) perpendicularly using a blade and then measuring it under a stereomicroscope to the nearest 0.01 mm. To measure the size of the gland, the gland was dissected from the gaster (thorax in bugs) using fine forceps, placed on a glass slide in a drop of water, and covered with a coverslip. The gland was approximately oval; thus the two perpendicular lengths of the gland were measured using an ocular ruler. We estimated gland area assuming a regular ellipsoid shape. We measured the sting length after pulling it from the abdomen using forceps. In each of these individuals we also measured the gaster/abdomen length, and thorax length. The ability to perform a communal attack (i.e. attack by several individuals) was taken from the literature and was assumed to be consistent throughout each genus. To estimate these traits for the remaining ant species that we failed to collect we used an interpolation based on proxy measures using 55 specimens: the size of both the poison/pygidial gland and sting length were regressed (using a linear model) from the size of the gaster, and cuticle thickness was regressed from the size of the thorax. These regression models (*Figure 8*) were then used to predict values for species we were unable to sample. The gland content varies among species of ants from different subfamilies: Formicinae ants possess formic acid (*O'Rourke, 1950*; *Stumper, 1952*), Dolichoderinae ants possess terpens, such as iridodial and dolichodial (*Cavill, 1960*; *Cavill and Hinterberger, 1960*), and Myrmeciinae ants possess indole (*Jackson et al., 1990*). The metathoracic scent gland in true bugs produces pungent pyrazines and alkenals (*Millar, 2005*). As the relative potency of these chemicals is unknown, we considered all these chemicals to be similarly noxious.

Various defensive traits measured here are not similarly effective against predators. Yet we assumed that their effect is similar because different predators adapted to deal with different defences of prey. We further assumed that the effect of these traits on predators is additive. So to quantify unpalatability ($u_j$) of the $j$th mimic we summed the $i$th trait values, which were scaled by their maximal value due to difference in range: $u_j = \sum_{j=1}^{n} \frac{x_{ij}}{\max(x_i)}$. We also computed a simple sum of defensive traits and found that the two measures are highly correlated (Pearson correlation, r = 0.72, p<0.0001). As the index of unpalatability includes more details we used it in all analyses. In order to test that the complex is dominated by moderately-defended species we used a Shapiro-Wilk test to test for a shape of a normal distribution of the unpalatability. To test whether the rings are composed of a similar range of unpalatable species, we used Bartlett test of variance homogeneity.

To test a hypothesis that unpalatability is related to the extent of gold colouration we used GLS to model the relationship between index of unpalatability and the absolute golden area (which is more ecologically relevant than relative area). The linear predictor included a quadratic term as the response was an area. A GLS model with Brownian motion model correlation structure was used to correct for the effect of phylogeny (*Pekár and Brabec, 2012*). To account for the fact that for some species, some defensive traits could not be measured (and were interpolated) we used weights that corresponded to the proportion of observed characteristics. Both variables were affected strongly by the body size so this measure needed to be included in the analysis. The body size had strong phylogenetic inertia (GLS, Pagel's λ = 1.08), thus by controlling for phylogeny the body size effect was included too, therefore use of the body size as a covariate was not necessary.

## Predation pressure by a community of predators

We investigated the diet of several predators on the campus of Macquarie University, North Ryde, Sydney, Australia. The campus in addition to buildings, contains a patch of native forest, several rocky streams, and open green space interspersed with trees where the predators could have been collected and observed which would be very difficult or even impossible in the pristine forest where the selection have taken place. The predators included noisy miners, lizards (skinks), and a number of arthropods that occur on the trunks and the immediate vicinity of Eucalyptus trees, where the mimics occur (see *Table 2*). We sampled arthropods from under the bark of more than 50 trees and selected those that were predatory. Arthropods were immediately placed into 99% pure ethanol and then in a freezer at −20°C. Co-existing predators were selected among the most abundant

species to represent three guilds according to their trophic strategy and prey detection modality: (1) visually-oriented euryphagous predators, (2) non-visually oriented ant-adverse predators, and (3) specialized ant-eating predators. We used the information of their gut content and data from the literature (*Jackson and Harding, 1982*; *Platnick, 2000*; *Higgins et al., 2001*; *McGinley et al., 2015*) to identify those that feed on prey related to the mimics. If more than 10% of individuals of a specific predator species were positive for primers designed for ants, the species was classified as ant-eating. Otherwise, the species was classified as euryphagous.

We collected 48 faecal samples from *Eulamprus quoyii* (Duméril et Bibron) (Eastern Water Skink, Scincidae, Squamata) and 48 samples from *Manorina melanocephala* Latham (Noisy Miner, Meliphagidae, Passeriformes) from across the Macquarie University campus during 2014 and 2015. These two predators were selected because they feed on arthropods, were most abundant among congeners and foraged on tree trunks. *Eulamprus quoyii* were collected from both rocky streams (human-altered) and forest-edge by noosing. Lizards typically defecated immediately upon capture and in some cases we gently massaged the abdomen to expel a faecal pellet. *Manorina melanocephala* faeces were collected at temporary feeding stations to avoid handling the birds. We lured birds to a large canvas (3 × 5 m) by placing small amounts of sweet cake near a water bowl and perches made from branches. The stations were monitored continuously. We sampled across a wide area in order to target specific family groups and the sampling was short enough to ensure that only droppings produced from previous meals were collected. We stored both the lizard and bird faecal samples individually in Eppendorf tubes in 80% ethanol in a −20°C freezer. This work was approved by the Animal Ethics Committee of Macquarie University (ARA 2014/031).

The gut of predators progressively degrades DNA through digestion (*Sint et al., 2011*). To account for the shortened DNA fragments we designed specific primers, which amplified fragments no longer than 300 bp (*Deagle et al., 2006*). This length is known to be a good compromise that increases the likelihood of amplification while still being long and variable enough to distinguish different prey taxa (*Herbert et al., 2003*; *Zeale et al., 2011*). A fragment of the COI gene was PCR amplified using LCO1490 (5′-GGTCAACAAATCATAAAGATATTGG-3′) and HCO2198 (5′-TAAAC TTCAGGGTGACCAAAAAATCA-3′) primers (*Folmer et al., 1994*) from representatives of potential prey found across the Macquarie University campus (Hemiptera: *Daerlac apicalis, D. cephalotes, D. nigricans*; Formicidae: *Myrmecia piliventris, M. tepperi, Camponotus aeneopilosus, Polyrhachis ammon, P. aurea, P. ornata, P. vermiculosa*; Araneae: *Myrmarachne luctuosa, M. erythrocephala*) and predators (Araneae: *Clubiona robusta* L. Koch, *Clubiona* sp.; *Euryopis umbilicata* L. Koch, *Euryopis* sp.; *Hemicloea* sp. 1, *Hemicloea* sp. 2; *Holoplatys planissima* (L. Koch); *Lampona murina* L. Koch; *Ocrisiona* sp.; *Sandalodes superbus* (Karsch); *Servaea incana* (Karsch)).

Each PCR (20 μL total volume) consisted of 5 μL of DNA, 1 μL of each primer (10 μM), 0.4 μL of 10 mM dNTP′s, 2.2 μL of 25 mM MgCl2, 2 μL of 10x PCR buffer, 1 μL of bovine albumin serum (BSA), 0.3 μL of Taq Polymerase (5 u/μL), and 7.1 μL of DNA-free water. PCR conditions were as follows: initial denaturation at 94°C for 3.5 min; 38 cycles of 94°C for 1 min, 44°C for 1 min (annealing temperature), 72°C for 1.5 min; a final extension at 72°C for 7 min. PCR products were detected by electrophoresis in 2% GoldView-stained agarose gels. Amplified products were purified using the

**Table 4.** List of primer pairs used to identify mimics in the gut or faeces of potential predators.

| Primer | Sequence | Fragment length (bp) | Amplified genera |
|---|---|---|---|
| DaerF | 5′- GAT CAA ATT TAT AAT AC −3′ | 204 | *Daerlac* |
| DaerR | 5′- TTC TGA TTA ATA AGG −3′ | | |
| FormF | 5′- GAT CAA ACY TTT AAY TC −3′ | 220 | *Camponotus, Myrmecia, Polyrhachis* |
| FormR | 5′- CCW GAT CCT TCA TTA ATA AA −3′ | | |
| MyrmF | 5′- ATT AGC TTC TAT TAT TG −3′ | 142 | *Myrmarachne* |
| MyrmR | 5′- TCT ATA GAA ATW CCT TCA G −3′ | | |
| BothmF | 5′- TCC TCA TGT TCA GGA ATA ATT AA −3′ | 215 | *Bothriomutilla* |
| BothmR | 5′- ATT AAG AGC ATA ATG GAT ATT GGG −3′ | | |

QIAquick PCR Purification Kit (Quiagen) and sequenced in both directions with BigDye Terminator v3.1 Sequence Kit (Applied Biosystems). Sequencing was carried out on an ABI Prism 3130 Genetic Analyzer (Applied Biosystems). Sequences were assembled in Sequencher 4.8 (Gene Codes Corporation, Ann Arbor, MI) and aligned using ClustalW (*Thompson et al., 1994*) implemented in MEGA 5.1 (*Tamura et al., 2011*). Based on sequence similarities/differences, group-specific primers, which would amplify DNA of mimics in the predator guts, were designed using Amplicon.b08 (*Jarman, 2004*). Four primer pairs were tested with all prey and predator specimens used in this study to ensure that the primers amplify DNA of the studied prey groups only: primers MyrmF and MyrmR amplified *Myrmarachne* spiders, DaerF and DaerR amplified *Daerlac* true bugs, FormF and FormR amplified *Camponotus*, *Polyrhachis* and *Myrmecia* ants, and BothmF and BothmR primers amplified *Bothriomutilla* mutillid wasps (for primer sequences see *Table 4*) and to reject cross-amplifications with the predators. The sequences of cytochrome c oxidase obtained from the studied predators and the mimics are deposited in GenBank (Accession numbers KX980351–KX980391).

DNA was extracted from 553 spider abdomens using DNeasy Blood and Tissue Kit (Qiagen) following the manufacturer's animal tissue protocol with a change in final step when only 50 µL of elution buffer was used. DNA from fecal samples (50 samples of skinks and 48 samples of noisy miner) was extracted using the DNeasy mericon Food Kit (Qiagen) following manufacturer's Small Fragment protocol. PCRs were performed using Multiplex PCR kit (Qiagen) under the following conditions: initial denaturation at 95°C for 15 min; 42 cycles of 94°C for 30 s, annealing temperature for 90 s (47.5°C when using Bothm primers, 43 and 45°C with the other primers), 72°C for 90 s; and a final extension at 72°C for 10 min. The reaction mixture total volume of 20 µL consisted of 10.6 µL of Multiplex PCR Master Mix, 1.8 µL of Q-Solution, 2.1 µL of RNase-free water, 0.5 µL of 10 µM forward and 0.5 µL of reverse primers, and 4.5 µL of DNA. PCR with Bothm primers was done separately whereas three remaining primer pairs were used in one multiplex reaction. PCR products were detected by electrophoresis in 2% GoodView-stained agarose gels. PCR was repeated for each sample and primer combination at least three times to exclude falsely negative results. In positive samples, PCR was repeated once again using primers modified with MID identifiers (i.e. multiplex identifiers, 10bp-long barcoding sequences). Each sample was thus marked with a unique combination of barcodes. PCR products were purified using a QIAquick PCR Purification Kit (Qiagen) according to the manufacturer's protocols. The concentration of each PCR product was measured using NanoDrop 8000 UV-Vis Spectrophotometer (Thermo Scientific, Waltman, MA) and assessed comparing to 50 bp ladder (Fermentas). Five µL of 50 µg/µL PCR products was pooled into the same sterile vial and sent for sequencing.

Sequencing library was prepared using Ion Plus Fragment Library Kit and Ion PI Template OT2 200 Kit v3 (Thermo Fisher Scientific). Sequencing on the Ion Proton System was provided by the SEQme company (Dobříš, Czech Republic). Sequencing data were processed using the Galaxy platform (https://usegalaxy.org/), BioEdit 7.2.5 (*Hall, 1999*) and MEGA 5.10. Reads were split according to their MIDs resulting in files corresponding to individual predators. Then, forward and reverse primers were removed and sequences were filtered according to their length (<120 or 180 bp) to remove dimers or too short reads. The sequences were collapsed and rare haplotypes (containing <2 identical sequences) were removed. Furthermore, sequences with stop codons and indels causing frameshifts were also removed. Remaining haplotypes were clustered into molecular operational taxonomic units (MOTU) using jMOTU 4.1 (*Jones et al., 2011*) with a 4 bp cut-off (corresponding to 3% sequence divergence). Each MOTU was compared to the GenBank database using megablast and BOLD database and to the sequences of mimics obtained within this study. NGS produced 11 068 904 informative reads from all predators of which 614,273 reads belonged to the mimics.

Availability of mimics to the predators was estimated by visual search on the trunks and below bark of 20 gum trees randomly selected on the campus. On each tree, abundance of mimics was surveyed by one person up to the height of 2 m from the ground for a period of 30 min on a sunny day. All mimics were collected, brought to the laboratory and identified to species. To test a hypothesis that the frequencies of captured mimics by all predators and those available at the microhabitat of predators are similar we used Chi-square test.

In order to estimate the attack rate exerted by the whole community of predators as precisely as possible, we had to take into account two important variables that affect the predation: (1) the relative frequency ($f_{ij}$) of positively screened $i$th predator individual for each $j$th mimic; (2) the capture

probability of araneophagous spider predator (*Lampona*), where body size is a strong predictor of the likelihood of predation. The capture probability was estimated for each spider predator and mimic (*Pekár et al., 2016*). This latter adjustment was necessary in order to take into account the fact that spiders could have captured juvenile *Myrmarachne* or *Daerlac* individuals, which are not golden mimics and could not be distinguished from adult specimens by molecular methods.

The functional model for the capture probability was obtained from data on eight spider species representing different guilds (*Nentwig and Wissel, 1986*). The logistic function was $\frac{1}{1+\exp(-1.99+2.11ratio_{ij})}$, where $ratio_{ij}$ is the mimic/predator body size ratio. The ratio was estimated by measuring the body size of ten collected specimens of each predator and mimic. Then the overall pressure by the community of predators was estimated as a sum of all attack rates by means of the killing value, $K_j$ (*Varley and Gradwell, 1960*): $K_j = \sum_{j=1}^{n} -\log(1-f_{ij})$. To test whether natural mortality caused by predation is related to unpalatability of mimics we used nonparametric regression as the response variable had non-standard distribution, namely generalized additive model (GAM) from the mgcv package (*Wood, 2011*). We regressed log-transformed $K$-values (in order to stay within positive bounds) against unpalatability index.

## Trials with selected predators

For predation trials we selected five species of mimics found on the Macquarie University campus. The five species represented a gradient of unpalatability (*Table 3*) within the mimetic complex. We included examples of the most unpalatable to the least palatable mimics: the ants *Polyrhachis ammon*, *Polyrhachis vermiculosa*, and *Camponotus aeneopilosus*, the true bug *Daerlac nigricans*, and the spider *Myrmarachne luctuosa*. Based on their unpalatability, the ants and bugs were predicted to be Müllerian mimics, while the spider was predicted to be a Batesian mimic. Unpalatability was re-estimated for these five species using ten individuals from each species. All attributes as measured previously were re-measured with the addition of a behavioural measure. The behavioural measure was the frequency of biting in the field following agitation with forceps. The addition of an extra measure adjusted the previous estimated unpalatability ($u_j$) but the relative relationship of unpalatability between species remained the same.

We used *Eulamprus quoyii* as our visually-oriented euryphagous vertebrate predator. At least 14 days prior to the experiment adult skinks were placed singly in a white container (70 × 50 × 40 cm) at constant temperature 26°C and L:D = 12:12 regime. They were fed regularly with three crickets (*Acheta domestica*) three times a week. The skinks were then starved five days prior to the predation trials. One day prior to the trial a layer of butter was applied to the sides of the white container to prevent arthropods from climbing the walls and escaping. At the start of a trial a lizard was left to settle down for a period of 5 min. A prey item was released into the container and the behaviour of the skink during the 5 min trial was recorded with a fixed video camera positioned above the container. Five minutes was deemed sufficient, as previous feeding trials with crickets have shown this species of lizard will attempt to capture prey within 1 min. If the potential prey survived it was removed, and the skink was left for 2 min to settle down. Another prey item was then released and so on until all five mimic prey types were used. As a non-mimic control, we used juveniles of the spider, *Badumna insignis* L. Koch, which was of a similar size to the other species but was not a mimic as it lacked golden colouration. At the end a cricket was released as a positive control. Results of trials were used in the analysis only if the cricket was captured.

The prey was offered to skinks using a constrained randomisation. There was randomisation within Müllerian mimics (ants) followed by randomisation within less defended mimics (*Daerlac*, *Myrmarachne*) and non-mimic prey (*Badumna*) because the predator must learn unpalatability by encountering the unpalatable models first. The learned aversion was necessary because skinks were collected in the wild a number of months prior to the trials and fed with crickets since then. We used 26 skinks in total. Ants, spiders and bugs were collected a few days prior to the trial. From the video footage we recorded whether prey was consumed or spat out.

The spider *Lampona murina* L. Koch was used as a nonvisually-oriented ant-adverse predator that will not capture ants but other spiders. Twenty-five late-instar juvenile Lampona spiders were collected from the bark of gum (*Eucalyptus* sp.) trees on the Macquarie University campus and kept individually in tubes (1 cm diameter, 10 cm long). Spiders were placed in a chamber set to 23°C and L:D = 16:8 regime. Spiders were fed with a clubionid spider once a week and provided with a

moistened piece of gauze. *Lampona* was not fed seven days prior to a trial. During a trial, an individual from the five mimetic prey species and one non-mimic spider (*Clubiona* sp.) was released into the Petri dish in a random order (diameter 6 cm). The prey was offered to *Lampona* in a random order. Induction of learned aversion was not necessary because the spiders were collected from the field only a week prior to the experiment. If the prey was not captured within 30 min we replaced it with a new one until one prey was captured. If the prey was captured, another prey was offered seven days later. In each trial we recorded capture frequency of *Lampona*.

Finally, we used the spider *Servaea incana* (Keyserling) as a specialised ant-eating predator. Twenty-seven adult individuals of *Servaea* spiders were collected from gum trees on the Macquarie University campus and placed singly in a Petri dish (8 cm diameter) and kept at 23°C and L:D = 16:8. Spiders were fed with small crickets once a week and provided with a moistened piece of gauze. We did not feed *Servaea* for five days prior to a trial. One individual of a mimic prey species was released into the dish. The prey was offered to *Servaea* in a random order. Induction of learned aversion was not necessary because spiders were collected from the field only a week prior to the experiment. If the prey was not captured within 5 min we replaced it with a new one. If the prey was captured another prey was offered five days later. In each trial we recorded whether the prey was captured as well as the behaviour of *Servaea* and prey.

To compare the frequency of capture among used prey types for each predator, relative frequencies of prey capture from *Eulamprus quoyi* or *Servaea* experiments were subjected to generalised estimating equations (GEE) from the geepack package with a binomial error structure due to repeated use of the same skink individuals with different prey. The exchangeable association structure was used (*Varley and Gradwell, 1960*). To test whether the post-attack reaction of the skinks was related to prey unpalatability we used generalized linear model with binomial error structure (GLM) of scores of the post-attack reaction of the skinks and the unpalatability index. The post-attack reaction was ranked as follows: 0 if the mimic was eaten, one if it was eaten but the skink cleaned its mouth, and two if the skink spat out the mimic. The ranks were then scaled to 0–1 range.

To compare the relative capture frequencies of *Lampona* on the prey species a Cochran Q test was used. To test whether overall mortality of prey is related to its unpalatability we used a general linear model (LM) of the sum of all probabilities of capture ($p_{ij}$) by the three predators and unpalatability index. The sum of capture rate was expressed as a killing value, $K_j$ using the formula: $K_j = \sum_{j=1}^{n} -\log(1 - p_{ij})$.

All statistical analyses were performed in R (*R Core Team, 2013*). For each model, diagnostic plots were used to assess model adequacy, such as variance homogeneity and distribution of residuals.

## Acknowledgements

We would like to thank very much P Lagos for help with the skink experiment; G Milledge, R Raven, D Smith, R White for providing access to spider or ant material; H Smith, G Corcobado, J Pekár, and M Pekár for help to collect spiders and ants in the field; G Cassis for identification of true bugs; R Kohout for access to ant pictures and advice on *Polyrhachis* taxonomy; D Noble for expert help while catching skinks and collecting faeces; D Birch (Macquarie University Microscopy Unit) for help with SEM photographs. The experiments with vertebrates were performed with the approval of the Macquarie University Animal ethics committee (AEC Reference No. 2013/033-3). We would like to thank the reviewers for their useful comments.

## Additional information

### Funding
No external funding was received for this work.

### Author contributions
SP, Formal analysis, Supervision, Investigation, Methodology, Writing—original draft, Writing—review and editing; LP, Formal analysis, Methodology, Writing—original draft; MWB, Resources,

Methodology, Writing—original draft; MJW, Methodology, Writing—original draft; MEH, Conceptualization, Writing—original draft, Writing—review and editing

### Author ORCIDs

Stano Pekár, http://orcid.org/0000-0002-0197-5040

### Ethics

Animal experimentation: The experiments with vertebrates were performed with approval of the Macquarie University Animal ethics committee (AEC Reference No. 2013/033-3).

## Additional files

### Major datasets

The following dataset was generated:

| Author(s) | Year | Dataset title | Dataset URL | Database, license, and accessibility information |
|---|---|---|---|---|
| Pekar S, Petrakova L, Bulbert MW, Whiting MJ, Herberstein ME | 2016 | Data from: The golden mimicry complex uses a spectrum of defences to deter a community of predators | http://dx.doi.org/10.5061/dryad.2c5t7 | Available at Dryad Digital Repository under a CC0 Public Domain Dedication |

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
