## [Decision Letter]

[Editors’ note: a previous version of this study was rejected after peer review, but the authors submitted for reconsideration. The first decision letter after peer review is shown below.]

Thank you for submitting your work entitled "The golden mimicry complex uses a spectrum of defences to deter a community of predators" for consideration by *eLife*. Your article has been reviewed by three peer reviewers, including Tom Reader (Reviewer #1), and the evaluation has been overseen by Merijn Kant as Reviewing Editor and Ian Baldwin and Detlef Weigel as Senior Editors.

Our decision has been reached after consultation between the reviewers. Based on these discussions and the individual reviews below, we regret to inform you that your work at this time will not be considered for publication in *eLife*. We like, however, the work in principle, and would very likely consider a thoroughly revised version again. We would treat it as a new submission, but try to retain the same reviewers. Note that this is in line with *eLife* policy, to invite revision directly only when the requested additional analyses can be reasonable accomplished within about two months. We estimate that this will take considerably longer in this case.

The referees agreed that this complex and rich system is novel and fascinating, but also noted that the sheer complexity of the system is also its weakness for analytical purposes. However, the referees are also confident that a presentation more firmly rooted into clear testable hypotheses to create a more logical question-answer structure will provide the insight, analytical coherency and perspective necessary for reaching a wide audience – like that of *eLife* – with a convincing story. The following points were prevalent during the discussion among the referees:

1) Justification and robustness of the morphometric index. You used a morphometric index for perceptual variables without justifying or critically evaluating the key assumptions used for the weighing. This makes it very difficult to understand how meaningful and robust this methodology is. Hence the weighing needs more explicit justification and it will be helpful to explore the robustness of the methodology by using alternative weighing of different variables – such as an alternative relative scaling of the different types of defenses – to get a sense of the validity of the claims.

2) A structure rooted in the testing of clear hypotheses. Overall you did not use a compelling question-answer structure and your analyses are often not founded on explicit logical assumptions or observations (e.g. supported by literature) so results can often not be compared with the expectations. You need to show the reader which observations indicate which type of ecological processes more clearly to separate beliefs from real insights. As you can see in the separate referee reports, the referees were also quite critical on many of the individual analyses. Designing the manuscript around a hypothesis-based framework will probably also provide a much more convincing basis for making and justifying explicit analytical decisions; for discussing these and for evaluating their robustness.

Taken together, the referees do believe that this fantastic and complex data set may encompass an exciting story but extracting this story in a convincing manner will require the logical and analytical structure of the manuscript to be revised from top to bottom.

Reviewer #1:

This exciting manuscript describes an extensive and novel mimicry complex. It is a very substantial study, employing a wide range of methods to unpick the biology of the mimicry complex. The work provides the research community with an excellent introduction to a new potential study system with which to address important evolutionary questions. I therefore think the study is novel, interesting to a wide audience, and important, and I applaud the authors for the huge effort required to collect the data. I especially like the use of next gen sequencing to identify real patterns of predation in the field (almost exactly what I was planning to do next year!). However, I am concerned that many of the conclusions are seriously flawed, and that some of the methods/rationale are not fully explained. I explain these concerns below.

1) Body size is a confounding variable in most of the analyses, and this diminishes the importance of many of the paper's conclusions. In the discussion, the authors briefly acknowledge the fact that body size might correlate with their measure of noxiousness. [Indeed, some of the components of noxiousness them ARE body size (subsection “Mimetic noxiousness”: missing values were taken from measures of body size).] The authors (rightly) point out that the absolute (as opposed to relative) size of defences such as spines may indeed matter to predators, so controlling for body size may reduce the information content of their noxiousness measure. However, body size matters to predators in many ways not directly related to the absolute size of defences: large prey have higher nutritional value and larger handling times, they may be unavailable to small predators because of anatomical constraints, they may be more likely to act as intraguild predators of small predators etc. So any relationship between predation and noxiousness (e.g. Figure 4 and Figure 5) may result from effects of body size other than those identified by the authors. The problems go beyond this: some of the interesting relationships identified in the results appear to be simple artefacts because of correlations with body size. For example, the relationship in Figure 4 shows an absolute measure of surface area (which obviously reflects body size) against noxiousness (which is known to reflect body size). Does this not just say that large animals have large surface areas? The authors have data on body size, so there is no difficultly in examining which relationships hold when body size effects are removed, either by expressing the constituent parts of noxiousness relative to body size, or by including body size as an alternative independent variable in the analysis. I agree with them that this will involve disposing of some of the useful biological information in the current noxiousness measure, and I would recommend displaying both kinds of analysis. If the relationships only hold when body size is not removed, the conclusion must be that any identified relationships involving noxiousness *may* (or may not) result from impacts of body size other than those related to the absolute size of spines, stings, venom glands etc.

2) The cluster analysis does not seem to be reported fully, and the groupings are not very obvious from Figure 3 (although the inset in Figure 1 helps a bit). How strong was the support for k = 10? Can you provide AIC scores for a range of models? The Manova doesn't really help in this respect, because any (arbitrary) partitioning of multivariate space would yield significantly different groupings. Which raw variables are important in distinguishing the clusters? How important is body size? I don't think it matters too much if the support for the clusters is weak: the existence of separate rings does not seem very important to the main points of interest in the paper. You might even ditch consideration of the rings altogether, or write a separate paper in which you consider the details.

3) Subsection “Mimicry and degree of noxiousness”. I don't see why the data suggest that "noxiousness promotes speciation", and Figure 4 seems only to confirm the idea that golden colours are (honestly) associated with defences: a genus with lots of (defended) golden species will have high mean noxiousness.

4) Some of the inferences and analysis are not very convincing.

A) Subsection “Predation pressure by a community of predators”: This analysis is not very convincing, because of small samples sizes, and the shape of the dataset. Which error structure does this model have? Given that the K value is 0 for most species, it seems unlikely that it is normal. Arguably it would be safer to use a Mann Whitney to compare the ranked k values for the least defended species (n = 2) with those for the more defended species.

B) The inferences made from the comparison of attack rates in Figure 6 are very dependent on the outcome for the non-mimics, for which there is no taxonomic replication (one species was chosen in each experiment). I don't find the pattern very convincing for this reason – is there a risk that the non-mimic was chosen (even subconsciously) because it was a particularly plausible prey species? Also, more specifically, since *Lampona* spiders were chosen because they are "ant-avoiders", does the pattern in the graph not just confirm that categorisation (i.e. the argument is circular)?

C) Subsection “Trials with selected predators” and Figure 5: This is a very weak test (n = 4), and the data show no trend at all, so I would avoid even implying some support for your prediction here.

D) Subsection “Trials with selected predators”. Prey type is confounded by order in the skink experiment. Any effect of defence/mimicry could be simply the result of some kind of habituation. Did you really need to rely on learned aversion during the experiment? Since these are wild-caught predators, could you not have assumed that the skinks already had learned (or innate) avoidance of aposematism? Why did you change to the more obvious randomised design with the spiders as predators (spiders learn too!)?

5) The colour analysis is not fully explained. I am confused about the colour measurement (Materials and methods section). How many taxa were photographed by the authors, and how many species were evaluated from the literature? Surely variation in lighting and camera sensors creates problems for measurements made from the literature? More importantly, how are the average RGB values used in the analysis? What are the 26 variables used in the cluster analysis (Statistical analyses)? Do these include measures of area as well as mean RGB values? Why do you use only "golden area" in Figure 4, and how was this measured/defined? The microscopy is not dealt with very systematically, either in the Materials and methods or the Results. What exactly did you measure/classify for each specimen, and what are we supposed to take from Figure 2?

6) I note that it is fairly unusual in the literature to use "noxiousness" to include physical defences such as spines, as well as chemical and behavioural defences. But the authors' choice of term does seem consistent with a strict dictionary definition.

Reviewer #2:

In this manuscript, Pekar and colleagues are describing the ecology of mimicry of a large assemblage of species displaying ant-like appearance in South-eastern Australia. The authors provide a wealth of ecological and morphological data to better understand the relationships between those species. The authors provide morphological measurements, estimates of noxiousness, and aim at confronting those to predatory data based on DNA sequencing from predator droppings and from behavioural experiments. The authors show that the rings are very speciose. They claim that the multiple ways of expressing defence, including distastefulness, venom and stingers, etc., contribute to equalling out the protection against a diverse set of predators, and lead to mutualistic interactions between the species participating in the mimicry ring.

The topic is interesting and understanding how highly speciose mimicry assemblages are formed is an important conundrum in the face of lab and theoretical results suggesting that mutualistic relationships are rare. I have not been convinced that the conclusions are unequivocally derived from the analysis of the data.

Firstly, the manuscript however is a composite account of all the kind of data obtainable on this mimicry ring, which together may hopefully help us form a better idea of the processes involved in this mimicry ring, but none of which is particularly convincing in its own right. The philosophy of this seems to be measuring as many things as we can, churning them into a multivariate analysis and taking this as a meaningful quantification of key ecological variables -morphology, defence/noxiousness, visual pattern, resemblance, etc. Unfortunately most of this is based on the postulate that if we measure many variables, they will together average out to something meaningful and hopefully not too far from the variation that is relevant to selection. I don't subscribe to this point of view. Resemblance is measured using a number of colour variables on different body parts, but it is impossible to understand how legitimate the absence of weighting is, in order for the composite measure to describe to morphological resemblance in a way that is relevant for predators. Similarly, noxiousness, an eminently perceptual notion, is "quantified" arbitrarily using size measurements of defence organs; although those don't necessarily correlate well with the perception of the noxious effects (tiny ants may have powerful venom, elongate mandibles may produce less painful a bite than short stubby mandibles, etc.). Generally, measurements are carried out on collection specimens or photographs, so I have less faith than the authors in the relevance of that to selection pressures in the wild. There is a hint of an effort to link perception with measurements, in Figure 4, but it is based on 6 data points (actually 4, since defenceless insects bring no information here).

Secondly, what are the questions, really? This manuscript is lacking clear hypotheses to be tested with real data. For each of the methods used, I would like to have a clear hypothesis spelled out, derived from an identifiable prediction taken form the literature, or perhaps from observation, and which the authors can test. The main conclusions of the manuscript, as it stands, lack support and perspective, and are generally muddled and based on "feelings", because the manuscript is not actually answering clear questions for which the study systems is well suited. As I detail below, the manuscript has many instances where a question emerges out of the blue, with no support from the literature, or where the logic is mysterious, or where the conclusions don't easily derive from the results.

For instance it is not clear why defence and signal strength should be correlated in the first place, otherwise Batesian mimics should not exist. The expectations regarding the consequence of having a diversity of defence mechanisms are absent, so we can't really evaluate whether or not the assemblage studied is indeed influenced by the distribution and structure of the defence variations. Similarly, the community of predators is indeed rich and diverse but the expectations of having multiple predators with different diets and affected in very different ways by prey defences are unclear, so again how this influences the evolution of this mimicry assemblage is impossible to tell. The "rings" the authors describe are analysed in terms of their geographic overlap but we don't know what the expectations are under different possible scenarios producing those rings, or what might underlie their clustered distribution in the morphological space. The authors ask whether mimicry should promote species richness in a clade, but why should mimetic clade be more speciose? Again the foundations of that conjecture are quite unclear. Those are just examples, and the net result is a manuscript where none of the data provide compelling insight into the mechanisms involved in the convergent evolution in the mimicry assemblage under study.

On another note, I praise the authors for the effort on investigating the diet of predators and establishing that certain prey are indeed part of the each predator's diet. I thought this was a probably the most useful contribution from this manuscript to the literature on mimicry ecology and something which should promote more research in the future (perhaps using higher throughput methods). However limited in power, the per predator (or per dropping) quantification is ok and gives a hint to the prevalence of certain prey in a predator's ecology. I would however be more careful with relating this to frequencies of prey in the field. Furthermore, the urban park setting is probably quite different from the predator's ecology in more pristine conditions where mimicry has evolved in those groups.

Reviewer #3:

In this paper, the authors identify 10 mimicry rings across 140 invertebrate taxa conduct a variety of tests (feeding trials and molecular gut content analyses) to determine how well-defended they are. Data collection and analyses are appropriate for the questions being addressed and I have no major issues with the paper.

[Editors’ note: what now follows is the decision letter after the authors submitted for further consideration.]

Thank you for submitting your article "The golden mimicry complex uses a spectrum of defences to deter a community of predators" for consideration by *eLife*. Your article has been reviewed by three peer reviewers, and the evaluation has been overseen by a Reviewing Editor and Ian Baldwin as the Senior Editor. The reviewers have opted to remain anonymous.

The reviewers have discussed the reviews with one another and the Reviewing Editor has drafted this decision to help you prepare a revised submission.

Summary:

Three new referees have evaluated your resubmitted manuscript. We agree that your paper is very interesting – albeit largely descriptive – and is based on an impressive dataset that reveals a fascinating system. Your study sheds valuable insights on some specific points: for example, that the color mimicry is attained through more than one mechanism so is convergent not just phylogenetically but in structural and pigmentary basis. Also, we appreciated the general conclusion that there seems to be no tighter association among sympatric species but that it is a continent-wide loose association. In addition we praise the approach and the usage of novel techniques to establish predation as well as the coherent presentation of the data. We are confident that this study will form the basis for more focused studies in the future.

Essential revisions:

Despite our largely positive opinion we still noted some issues that remain unsatisfactorily answered and we suggest a couple of additions. It will largely come down to reducing the strength of some claims and using more careful language. Please take the following issues seriously and note that *eLife* does not have any space limitations:

1) Your manuscript leaves too much the impression that mimicry studies started in the 21st century. In particular there were the huge studies of bird gut contents by McAtee in the early 20th century, used by him to deny the efficacy of mimetic protection. We propose a discussion of this point in subsection “Predation pressure by a community of predators”;

2) In relation to point 1 and subsection “Predation pressure by a community of predators”, we did not understand how the frequency of captured mimics of 13 species can be no different from their abundance in the environment, and yet only 3 of them were in fact captured at all? Some data and clarification is needed here;

3) We suggest you to refer and discuss the paper of Nicholson (1927, Australian Zoologist 5: 10-104), who was the first to suggest the same message as your study (on the basis of Australian insects), namely that mimicry extends to entire complexes of organisms, and is a phenomenon far more important and widespread than is appreciated even today;

4) You defined 'quasi-Batesian mimicry' as 'less defended mimics' (Introduction section), but to our understanding Speed 1993 defined it as 'unequally defended prey' under the assumption that less unpalatable mimics could raise the probability of attack on their more unpalatable models thereby (possibly) leading to the evolution of a polymorphism in the defended mimic. So it would be more accurate to say here that: "discrepancy in protection between two mimetic species may dilute the protection of a better defended species";

5) We feel you should acknowledge (in the Discussion) more explicitly that your study is a composite account of all the kind of variation of this mimicry ring that, however, does not allow for robust conclusions on the evolution of visual mimicry among the studied species i.e. how the different components of 'resemblance' and 'noxiousness' were selected for in different taxa, with different levels of association in different geographical areas. For example, there is no clear-cut difference in attack behavior of visual and non-visual predators and there is no significant difference in the proportion of 'mimetic' prey in the diet as compared to their abundance in the wild (sequencing data). Overall you need to indicate in your discussion more explicitly that your formal demonstration of visual mimicry among the different species is rather limited;

6) With respect to point 5 we also would like to encourage you to make some suggestions for how to unpick all this fascinating detail in the future;

7) The conclusions derived from the skink results need to be toned down since prey type is confounded by order in this experiment. Any effect of defence/mimicry could be simply the result of some kind of habituation. This needs to be indicated clearly in the discussion;

8) We noticed that there is some ambiguity in what is being inferred from the cross-species comparisons. For example in the Introduction section you write that warning signals and distastefulness are expected to evolve together and in subsection “Mimicry and degree of unpalatability” you report on a positive correlation between the golden coloration and unpalatability. However, the theory of honest signaling only predicts a correlation between signal intensity and unpalatability within a species. A correlation across species is interesting as an observed fact needing explanation, but it doesn't support any particular theory for the evolution of aposematic signals (we are aware of published papers which use cross-species comparisons to 'test' honest signaling theories, but they are wrong to do so). Hence these results need a different framing;

9) We ask you express more explicit caution with respect to the ranked intensities (subsection “Description of the mimetic complex”) since, as we understand it, the photographs were standardized to the same white point, but were not linearized (RGB pixel values are unlikely to be linearly related to radiance). This means that intensities may be rankable, but numerical differences can't be relied upon. Although this does not affect analyses of golden patch area, any RGB measures of signal intensity must have a large 'caution' attached to them. This could easily be addressed in the Discussion;

10) The hypothetical mimicry relying on non-visual component is interesting and is suggested by the behavior of non-visual predators but the data presented here do not formally demonstrate it. This should be indicated more explicitly in the discussion.

11) Finally the contribution of the mildly defended species to the overall protection gained by all members of the mimetic community is also not formally demonstrated. Also this should be acknowledged more explicitly in the discussion.

---

## [Author Response]

[Editors’ note: the author responses to the first round of peer review follow.]

*[…]Reviewer #1:*

*This exciting manuscript describes an extensive and novel mimicry complex. It is a very substantial study, employing a wide range of methods to unpick the biology of the mimicry complex. The work provides the research community with an excellent introduction to a new potential study system with which to address important evolutionary questions. I therefore think the study is novel, interesting to a wide audience, and important, and I applaud the authors for the huge effort required to collect the data. I especially like the use of next gen sequencing to identify real patterns of predation in the field (almost exactly what I was planning to do next year!). However, I am concerned that many of the conclusions are seriously flawed, and that some of the methods/rationale are not fully explained. I explain these concerns below.*

*1) Body size is a confounding variable in most of the analyses, and this diminishes the importance of many of the paper's conclusions. In the discussion, the authors briefly acknowledge the fact that body size might correlate with their measure of noxiousness. [Indeed, some of the components of noxiousness them ARE body size (subsection “Mimetic noxiousness”: missing values were taken from measures of body size).] The authors (rightly) point out that the absolute (as opposed to relative) size of defences such as spines may indeed matter to predators, so controlling for body size may reduce the information content of their noxiousness measure. However, body size matters to predators in many ways not directly related to the absolute size of defences: large prey have higher nutritional value and larger handling times, they may be unavailable to small predators because of anatomical constraints, they may be more likely to act as intraguild predators of small predators etc. So any relationship between predation and noxiousness (e.g. Figure 4 and Figure 5) may result from effects of body size other than those identified by the authors. The problems go beyond this: some of the interesting relationships identified in the results appear to be simple artefacts because of correlations with body size. For example, the relationship in Figure 4 shows an absolute measure of surface area (which obviously reflects body size) against noxiousness (which is known to reflect body size). Does this not just say that large animals have large surface areas? The authors have data on body size, so there is no difficultly in examining which relationships hold when body size effects are removed, either by expressing the constituent parts of noxiousness relative to body size, or by including body size as an alternative independent variable in the analysis. I agree with them that this will involve disposing of some of the useful biological information in the current noxiousness measure, and I would recommend displaying both kinds of analysis. If the relationships only hold when body size is not removed, the conclusion must be that any identified relationships involving noxiousness *may* (or may not) result from impacts of body size other than those related to the absolute size of spines, stings, venom glands etc.*

We are aware of the confounding effects. Firstly, we replaced the term noxiousness by unpalatability throughout the manuscript as it better describes the fact that mimics are also protected by large body size, which is particularly important for arthropod predators. We explain this in the Discussion. Both variables, unpalatability and absolute golden area, were affected strongly by the body size so this measure needs to be included in the analysis. We re-run the analyses with body size as a covariate and the effects were still significant as without the covariate but with the phylogenetic correlation structure. The body size had strong phylogenetic inertia (Pagel’s λ = 1.08), thus by controlling for phylogeny the body size effect was included too, therefore use of the body size as a covariate was not necessary.

*2) The cluster analysis does not seem to be reported fully, and the groupings are not very obvious from Figure 3 (although the inset in Figure 1 helps a bit). How strong was the support for k = 10? Can you provide AIC scores for a range of models? The Manova doesn't really help in this respect, because any (arbitrary) partitioning of multivariate space would yield significantly different groupings. Which raw variables are important in distinguishing the clusters? How important is body size? I don't think it matters too much if the support for the clusters is weak: the existence of separate rings does not seem very important to the main points of interest in the paper. You might even ditch consideration of the rings altogether, or write a separate paper in which you consider the details.*

We removed MANOVA and provided plot of the classification criterion used. AIC is not available to this function. But a similar measure is a Simple structure index. The support for rings is quite weak so we reduced their importance.

*3) Subsection “Mimicry and degree of noxiousness”. I don't see why the data suggest that "noxiousness promotes speciation", and Figure 4 seems only to confirm the idea that golden colours are (honestly) associated with defences: a genus with lots of (defended) golden species will have high mean noxiousness.*

This statement and the corresponding analysis were omitted.

4) Some of the inferences and analysis are not very convincing.

*A) Subsection “Predation pressure by a community of predators”: This analysis is not very convincing, because of small samples sizes, and the shape of the dataset. Which error structure does this model have? Given that the K value is 0 for most species, it seems unlikely that it is normal. Arguably it would be safer to use a Mann Whitney to compare the ranked k values for the least defended species (n = 2) with those for the more defended species.*

The sample sizes are not small (altogether more than 500 inds were analysed) but the capture rate is very small. Yet, it reflects the real situation. The linear model was replaced by nonparametric regression that does not rely on distribution assumptions. The results are, however, similar to the linear model.

*B) The inferences made from the comparison of attack rates in Figure 6 are very dependent on the outcome for the non-mimics, for which there is no taxonomic replication (one species was chosen in each experiment). I don't find the pattern very convincing for this reason – is there a risk that the non-mimic was chosen (even subconsciously) because it was a particularly plausible prey species? Also, more specifically, since Lampona spiders were chosen because they are "ant-avoiders", does the pattern in the graph not just confirm that categorisation (i.e. the argument is circular)?*

In fact, two nonmimetic species were used. One was used in the experiment, and the other (cricket, clubionid spiders) was used to feed the predators between trials. The non-mimetic species was chosen to meet the following criteria: abundant in the environment of mimics, be of similar size to the mimics, have no apparent colouration, potential prey of putative predators. *Lampona* was expected to be an ant-avoider. It is a specialised spider-eating predator and thus should catch spiders. Yet *Myrmarachne* (a spider) that imitates ants has not been so frequently captured.

*C) Subsection “Trials with selected predators” and Figure 5: This is a very weak test (n = 4), and the data show no trend at all, so I would avoid even implying some support for your prediction here.*

Rephrased as suggested.

*D) Subsection “Trials with selected predators”. Prey type is confounded by order in the skink experiment. Any effect of defence/mimicry could be simply the result of some kind of habituation. Did you really need to rely on learned aversion during the experiment? Since these are wild-caught predators, could you not have assumed that the skinks already had learned (or innate) avoidance of aposematism? Why did you change to the more obvious randomised design with the spiders as predators (spiders learn too!)?*

The learned aversion was necessary because skinks were collected in the wild a number of months prior to the trials and fed with crickets since then. Induction of learned aversion was not necessary in case of spiders because they were collected from the field only a week prior to the experiment.

*5) The colour analysis is not fully explained. I am confused about the colour measurement (Materials and methods section). How many taxa were photographed by the authors, and how many species were evaluated from the literature? Surely variation in lighting and camera sensors creates problems for measurements made from the literature? More importantly, how are the average RGB values used in the analysis? What are the 26 variables used in the cluster analysis (Statistical analyses)? Do these include measures of area as well as mean RGB values? Why do you use only "golden area" in Figure 4, and how was this measured/defined? The microscopy is not dealt with very systematically, either in the methods or the results. What exactly did you measure/classify for each specimen, and what are we supposed to take from Figure 2?*

We added more details to the description of colour analysis and microscopy.

*6) I note that it is fairly unusual in the literature to use "noxiousness" to include physical defences such as spines, as well as chemical and behavioural defences. But the authors' choice of term does seem consistent with a strict dictionary definition.*

The term was replaced by unpalatability.

*Reviewer #2:*

*In this manuscript, Pekar and colleagues are describing the ecology of mimicry of a large assemblage of species displaying ant-like appearance in South-eastern Australia. The authors provide a wealth of ecological and morphological data to better understand the relationships between those species. The authors provide morphological measurements, estimates of noxiousness, and aim at confronting those to predatory data based on DNA sequencing from predator droppings and from behavioural experiments. The authors show that the rings are very speciose. They claim that the multiple ways of expressing defence, including distastefulness, venom and stingers, etc., contribute to equalling out the protection against a diverse set of predators, and lead to mutualistic interactions between the species participating in the mimicry ring.*

*The topic is interesting and understanding how highly speciose mimicry assemblages are formed is an important conundrum in the face of lab and theoretical results suggesting that mutualistic relationships are rare. I have not been convinced that the conclusions are unequivocally derived from the analysis of the data.*

*Firstly, the manuscript however is a composite account of all the kind of data obtainable on this mimicry ring, which together may hopefully help us form a better idea of the processes involved in this mimicry ring, but none of which is particularly convincing in its own right. The philosophy of this seems to be measuring as many things as we can, churning them into a multivariate analysis and taking this as a meaningful quantification of key ecological variables -morphology, defence/noxiousness, visual pattern, resemblance, etc. Unfortunately most of this is based on the postulate that if we measure many variables, they will together average out to something meaningful and hopefully not too far from the variation that is relevant to selection. I don't subscribe to this point of view. Resemblance is measured using a number of colour variables on different body parts, but it is impossible to understand how legitimate the absence of weighting is, in order for the composite measure to describe to morphological resemblance in a way that is relevant for predators.*

We have used weighting in the multivariate analysis but forgot to include it in the Methods. Now the information is included.

*Similarly, noxiousness, an eminently perceptual notion, is "quantified" arbitrarily using size measurements of defence organs; although those don't necessarily correlate well with the perception of the noxious effects (tiny ants may have powerful venom, elongate mandibles may produce less painful a bite than short stubby mandibles, etc.). Generally, measurements are carried out on collection specimens or photographs, so I have less faith than the authors in the relevance of that to selection pressures in the wild. There is a hint of an effort to link perception with measurements, in Figure 4, but it is based on 6 data points (actually 4, since defenceless insects bring no information here).*

The quantification was not arbitrary. We used to our knowledge the best information available. We acknowledge in the Discussion that effect of defences is species-specific. There are no defenceless species (such species cannot exist as it would have been immediately consumed by predators). Thus even species with weak defences add to our information about the relationship. And the Figure 4 in fact provides direct support to the relevance of the index.

*Secondly, what are the questions, really? This manuscript is lacking clear hypotheses to be tested with real data. For each of the methods used, I would like to have a clear hypothesis spelled out, derived from an identifiable prediction taken form the literature, or perhaps from observation, and which the authors can test. The main conclusions of the manuscript, as it stands, lack support and perspective, and are generally muddled and based on "feelings", because the manuscript is not actually answering clear questions for which the study systems is well suited. As I detail below, the manuscript has many instances where a question emerges out of the blue, with no support from the literature, or where the logic is mysterious, or where the conclusions don't easily derive from the results.*

We have discovered the complex so it really came out of the blue. However, the Introduction has been rephrased to pose clear questions that the paper tests.

*For instance it is not clear why defence and signal strength should be correlated in the first place, otherwise Batesian mimics should not exist. The expectations regarding the consequence of having a diversity of defence mechanisms are absent, so we can't really evaluate whether or not the assemblage studied is indeed influenced by the distribution and structure of the defence variations. Similarly, the community of predators is indeed rich and diverse but the expectations of having multiple predators with different diets and affected in very different ways by prey defences are unclear, so again how this influences the evolution of this mimicry assemblage is impossible to tell. The "rings" the authors describe are analysed in terms of their geographic overlap but we don't know what the expectations are under different possible scenarios producing those rings, or what might underlie their clustered distribution in the morphological space. The authors ask whether mimicry should promote species richness in a clade, but why should mimetic clade be more speciose? Again the foundations of that conjecture are quite unclear. Those are just examples, and the net result is a manuscript where none of the data provide compelling insight into the mechanisms involved in the convergent evolution in the mimicry assemblage under study.*

We extended the end of Introduction section to include clear hypothesis that are tested in the Results section.

*On another note, I praise the authors for the effort on investigating the diet of predators and establishing that certain prey are indeed part of the each predator's diet. I thought this was a probably the most useful contribution from this manuscript to the literature on mimicry ecology and something which should promote more research in the future (perhaps using higher throughput methods). However limited in power, the per predator (or per dropping) quantification is ok and gives a hint to the prevalence of certain prey in a predator's ecology. I would however be more careful with relating this to frequencies of prey in the field. Furthermore, the urban park setting is probably quite different from the predator's ecology in more pristine conditions where mimicry has evolved in those groups.*

The urban park is different. However, such study could not be performed out in the pristine nature. Firstly, because of dangerous venomous animals that dwell in the litter and also because of logistic problems – for example, collection of bird poo would be impossible in a pristine forest.

[Editors' note: the author responses to the re-review follow.]

*Essential revisions:*

*Despite our largely positive opinion we still noted some issues that remain unsatisfactorily answered and we suggest a couple of additions. It will largely come down to reducing the strength of some claims and using more careful language. Please take the following issues seriously and note that eLife does not have any space limitations:*

*1) Your manuscript leaves too much the impression that mimicry studies started in the 21st century. In particular there were the huge studies of bird gut contents by McAtee in the early 20th century, used by him to deny the efficacy of mimetic protection. We propose a discussion of this point in subsection “Predation pressure by a community of predators”;*

We think such information should appear earlier. So we rephrased one sentence in the Introduction and added citations of two McAtte’s papers.

*2) In relation to point 1 and subsection “Predation pressure by a community of predators”, we did not understand how the frequency of captured mimics of 13 species can be no different from their abundance in the environment, and yet only 3 of them were in fact captured at all? Some data and clarification is needed here;*

This is because mimics were also very rare among available prey as is now explained in the Results section.

*3) We suggest you to refer and discuss the paper of Nicholson (1927, Australian Zoologist 5: 10-104), who was the first to suggest the same message as your study (on the basis of Australian insects), namely that mimicry extends to entire complexes of organisms, and is a phenomenon far more important and widespread than is appreciated even today;*

Many thanks for this suggestion. We have overlooked this important paper. We added a sentence referring to this paper in the beginning of Discussion.

*4) You defined 'quasi-Batesian mimicry' as 'less defended mimics' (Introduction section), but to our understanding Speed 1993 defined it as 'unequally defended prey' under the assumption that less unpalatable mimics could raise the probability of attack on their more unpalatable models thereby (possibly) leading to the evolution of a polymorphism in the defended mimic. So it would be more accurate to say here that: "discrepancy in protection between two mimetic species may dilute the protection of a better defended species";*

We rephrased the sentence as suggested.

*5) We feel you should acknowledge (in the discussion) more explicitly that your study is a composite account of all the kind of variation of this mimicry ring that, however, does not allow for robust conclusions on the evolution of visual mimicry among the studied species i.e. how the different components of 'resemblance' and 'noxiousness' were selected for in different taxa, with different levels of association in different geographical areas. For example, there is no clear-cut difference in attack behavior of visual and non-visual predators and there is no significant difference in the proportion of 'mimetic' prey in the diet as compared to their abundance in the wild (sequencing data). Overall you need to indicate in your discussion more explicitly that your formal demonstration of visual mimicry among the different species is rather limited;*

We added a new paragraph towards the end of Discussion section acknowledging all these concerns.

*6) With respect to point 5 we also would like to encourage you to make some suggestions for how to unpick all this fascinating detail in the future;*

This information is provided in the same paragraph as above concerns.

*7) The conclusions derived from the skink results need to be toned down since prey type is confounded by order in this experiment. Any effect of defence/mimicry could be simply the result of some kind of habituation. This needs to be indicated clearly in the discussion;*

This is discussed in another new paragraph.

*8) We noticed that there is some ambiguity in what is being inferred from the cross-species comparisons. For example in the Introduction section you write that warning signals and distastefulness are expected to evolve together and in subsection “Mimicry and degree of unpalatability” you report on a positive correlation between the golden coloration and unpalatability. However, the theory of honest signaling only predicts a correlation between signal intensity and unpalatability within a species. A correlation across species is interesting as an observed fact needing explanation, but it doesn't support any particular theory for the evolution of aposematic signals (we are aware of published papers which use cross-species comparisons to 'test' honest signaling theories, but they are wrong to do so). Hence these results need a different framing;*

Thanks for this hint. We reformulated sentences in the Introduction and Discussion sections to reflect this difference (intra- vs. inter- specific relationship) and added references for the interspecific comparison.

*9) We ask you express more explicit caution with respect to the ranked intensities (subsection “Description of the mimetic complex”) since, as we understand it, the photographs were standardized to the same white point, but were not linearized (RGB pixel values are unlikely to be linearly related to radiance). This means that intensities may be rankable, but numerical differences can't be relied upon. Although this does not affect analyses of golden patch area, any RGB measures of signal intensity must have a large 'caution' attached to them. This could easily be addressed in the Discussion;*

This is discussed in the new paragraph (as above).

*10) The hypothetical mimicry relying on non-visual component is interesting and is suggested by the behavior of non-visual predators but the data presented here do not formally demonstrate it. This should be indicated more explicitly in the discussion.*

This is discussed in the new paragraph (as above).

*11) Finally the contribution of the mildly defended species to the overall protection gained by all members of the mimetic community is also not formally demonstrated. Also this should be acknowledged more explicitly in the discussion.*

This is discussed in the new paragraph (as above).